# Onco-Ontogeny of Squamous Cell Cancer of the First Pharyngeal Arch Derivatives

**DOI:** 10.3390/ijms25189979

**Published:** 2024-09-16

**Authors:** Daniel Sat-Muñoz, Luz-Ma.-Adriana Balderas-Peña, Eduardo Gómez-Sánchez, Brenda-Eugenia Martínez-Herrera, Benjamín Trujillo-Hernández, Luis-Aarón Quiroga-Morales, Mario Salazar-Páramo, Ingrid-Patricia Dávalos-Rodríguez, Carlos M. Nuño-Guzmán, Martha-Cecilia Velázquez-Flores, Miguel-Ricardo Ochoa-Plascencia, María-Ivette Muciño-Hernández, Mario-Alberto Isiordia-Espinoza, Mario-Alberto Mireles-Ramírez, Eduardo Hernández-Salazar

**Affiliations:** 1Departamento de Morfología, Centro Universitario de Ciencis de la Salud, Universidad de Guadalajara, Guadalajara 44340, Mexico; c.velazquez@academicos.udg.mx; 2Cuerpo Académico UDG-CA-874, Ciencias Morfológicas en el Diagnóstico y Tratamiento de la Enfermedad, Centro Universitario de Ciencias de la Salud, Universidad de Guadalajara, Guadalajara 44340, Mexico; eduardo.gsanchez@academicos.udg.mx (E.G.-S.); miguel.oplascencia@academicos.udg.mx (M.-R.O.-P.); ivette.mucino@academicos.udg.mx (M.-I.M.-H.); 3Unidad Médica de Alta Especialidad (UMAE), Departamento Clínico de Cirugía Oncológica, Hospital de Especialidades (HE), Centro Médico Nacional de Occidente (CMNO), Instituto Mexicano del Seguro Social (IMSS), Guadalajara 44340, Mexico; 4Comité de Tumores de Cabeza y Cuello, Unidad Médica de Alta Especialidad (UMAE), Hospital de Especialidades (HE), Centro Médico Nacional de Occidente (CMNO), Instituto Mexicano del Seguro Social (IMSS), Guadalajara 44340, Mexico; 5Unidad de Investigación Biomédica 02, Unidad Médica de Alta Especialidad (UMAE), Hospital de Especialidades (HE), Centro Médico Nacional de Occidente (CMNO), Instituto Mexicano del Seguro Social (IMSS), Guadalajara 44340, Mexico; 6División de Disciplinas Clínicas, Centro Universitario de Ciencias de la Salud, Universidad de Guadalajara, Guadalajara 44340, Mexico; carlosnunoguzman@hotmail.com; 7Departamento de Nutrición y Dietética, Hospital General de Zona #1, Instituto Mexicano del Seguro Social, OOAD Aguascalientes, Boulevard José María Chavez #1202, Fracc, Lindavista, Aguascalientes 20270, Mexico; bren.mtzh16@gmail.com; 8Posgrado en Ciencias Médicas, Universidad de Colima, Colima 28040, Mexico; trujillobenjamin@hotmail.com; 9Unidad Académica de Ciencias de la Salud, Clínica de Rehabilitación y Alto Rendimiento ESPORTIVA, Universidad Autónoma de Guadalajara, Zapopan 45129, Mexico; luisquiroga@hotmail.com; 10Departamento de Fisiología, Centro Universitario de Ciencias de la Salud, Universidad de Guadalajara, Academia de Inmunología, Guadalajara 44340, Mexico; mario.sparamo@academicos.udg.mx (M.S.-P.); ingriddavalos@hotmail.com (I.-P.D.-R.); 11Departamento de Biología Molecular y Genómica, División de Genética, Centro de Investigación Biomédica de Occidente, Instituto Mexicano del Seguro Social. Guadalajara 44340, Mexico; 12Departamento Clínico de Cirugía General, Unidad Médica de Alta Especialidad (UMAE), Hospital de Especialidades, Centro Médico Nacional de Occidente, Instituto Mexicano del Seguro Social, Guadalajara 44340, Mexico; 13Unidad Médica de Alta Especialidad (UMAE), Departamento Clínico de Anestesiología, División de Cirugía, Hospital de Especialidades, Centro Médico Nacional de Occidente, Instituto Mexicano del Seguro Social, Guadalajara 44340, Mexico; 14Departamento de Clínicas, División de Ciencias Biomédicas, Centro Universitario de los Altos, Instituto de Investigación en Ciencias Médicas, Cuerpo Académico Terapéutica y Biología Molecular (UDG-CA-973), Universidad de Guadalajara, Tepatitlán de Morelos 47620, Mexico; mario.isiordia162@yahoo.com; 15División de Investigación en Salud, UMAE, Hospital de Especialidades, Centro Médico Nacional de Occidente, Instituto Mexicano del Seguro Social, Guadalajara 44340, Mexico; dr_mireles@hotmail.com; 16Departamento de Admisión Médica Continua, UMAE Hospital de Especialidades, Centro Médico Nacional de Occidente, Instituto Mexicano del Seguro Social, Guadalajara 44340, Mexico; ehs_internista@yahoo.com.mx

**Keywords:** onco-ontogeny, head and neck squamous cell carcinoma, notch, HOX, BMP

## Abstract

Head and neck squamous cell carcinoma (H&NSCC) is an anatomic, biological, and genetic complex disease. It involves more than 1000 genes implied in its oncogenesis; for this review, we limit our search and description to the genes implied in the onco-ontogeny of the derivates from the first pharyngeal arch during embryo development. They can be grouped as transcription factors and signaling molecules (that act as growth factors that bind to receptors). Finally, we propose the term embryo-oncogenesis to refer to the activation, reactivation, and use of the genes involved in the embryo’s development during the oncogenesis or malignant tumor invasion and metastasis events as part of an onco-ontogenic inverse process.

## 1. Introduction

Cancer is a disease characterized by uncontrolled cell division. The processes that lead to division, growth, invasion, and spread are tightly regulated at the molecular level. The aging population is experiencing higher rates of carcinoma, including head and neck squamous cell carcinoma (H&NSCC), which is strongly linked to age, as mentioned in data published by Barsouk, Sung, and Johnson, among others [1,2,3,4].

H&NSCC refers to cancers that develop in the upper aerodigestive epithelium, which includes the paranasal sinuses, nasal cavity, oral cavity, pharynx, and larynx. Most of these cancers are squamous cell carcinomas. They are associated with factors such as tobacco exposure, excessive alcohol consumption, and human papillomavirus (HPV), particularly in oropharyngeal and laryngeal cancers [1,5], and Epstein–Barr virus (EBV) in nasopharyngeal carcinoma, as predictors of overall survival and treatment response [6,7,8].

In 2018, head and neck cancers were the seventh most common carcinoma worldwide, responsible for 890,000 new cases and 450,000 deaths. By 2020, H&NSCC had become the sixth most frequent carcinoma globally, with an incidence of 930,000 and 460,000 related deaths [9]. Projections suggest that the incidence will rise to 1,080,000 new cases annually by 2030 [1]. In the United States, it accounts for 3% of all cancers (51,540 new cases) and just over 1.5% of all carcinoma deaths (10,030 deaths) [10]. Typically, the disease is diagnosed in older patients [4]. The global incidence is expected to decline gradually, partly due to reduced tobacco use [11]. 

H&NSCC is a diverse disease with different biological behavior patterns [12]. Around two-thirds of patients with H&NSCC have advanced-stage disease, which is characterized by large tumors with local invasion and metastasis to the local, regional nodes, or both. Concerning distant metastasis, a recently published paper described how immunosuppression, nodal metastasis over 6 mm, more than five nodal metastasis, and bilateral disease can accurately predict a high risk of distant metastatic disease [13], and 70% of treatment failures are related to the presence of distant metastasis [14]. The 5-year overall survival rate for these patients is less than 50%. At the time of diagnosis, only 30–40% of patients present with clinical stage I or II of the disease [5]. 

The treatment of these types of cancers is very complex due to the anatomical region involved, requiring a multidisciplinary approach involving medical oncologists, surgical oncologists, maxillofacial surgeons, radiation oncologists, plastic surgeons, nutritionists, psycho-oncologists, and neurosurgeons [15]. This means that the treatment needs to be multimodal. However, it is complicated by the fact that most patients are around 60 years old on average, which results in other associated pathologies such as hypertension, diabetes, and malnutrition [16,17,18]. These additional health issues could make a patient unsuitable for treatment.

According to the analysis tool, it has been reported that between 50 and 100 genes are mutated in H&NSCC, which could be considered carcinoma-driver genes [19,20]. These genes have a close relationship with the aging process during carcinogenesis.

The biological pattern of H&NSCC is closely linked to HPV infection in sites such as the oropharynx and larynx. This has led the Carcinoma Genome Atlas to classify H&NSCC as HPV negative (HPV−) and HPV positive (HPV+) to determine whether there is a difference in gene expression related to HPV infection [21,22].

In 2022, Habib I et al. conducted a study on patients’ genome data with H&NSCC to identify gene expression patterns in these cancers. They described the gene expression pattern and patients’ survival for ten genes: *PRAME*, *MAGEC2*, *MAGEA12*, *LHX1*, *MAGEA3*, *CSAG1*, *MAGEA6*, *LCE6A*, *LCE2D*, and *LCE2C*. Among these genes, *LHX1* is known to be involved in early embryo head and neck development [21]. These findings suggest the potential role of genes related to embryonic development in developing H&NSCC.

Due to the complex nature of the developmental biology of the head and neck, which involves numerous genes, both known and yet to be discovered, we will focus our review on the genes involved in the cancerous development of the structures derived from the first pharyngeal arch that may play a role in the development of H&NSCC.

## 2. Parallelisms between the Development of the Embryo and Malignant Cells

Cancer genesis appears to be a highly chaotic process to the naked eye. Still, this process is highly controlled at a molecular level, and a combination of multiple genetic and epigenetic factors is required to develop a malignant tumor successfully.

In the early nineteenth century, Virchow hypothesized that malignant tumors arise from embryonic cells or cells resembling those in embryos [23,24].

It has been 70 years since the initial systematic experimental work on teratomas and teratocarcinomas [25]. And there have been many attempts to compare embryonic development with malignant transformation. This is probably because, as we have seen, there exist a lot of similarities in the behavior of normal cells during embryogenesis (ESCs: embryonic stem cells) and tumor cells during carcinogenesis (progression, invasion, and metastasis). The presence of gene mutations in the genetic pathways activated during embryonic development is frequent in malignant cell development, as observed during the migration of neural crest cells (NCCs) and invasive malignant cells during invasion and metastasis [26,27].

In 1829, Joseph Récamier suggested that primitive germ cells might give rise to some tumors; in 1874, Francesco Durante proposed an embryonic rest theory in which he envisioned the possible routes to carcinoma through embryonic intermediates. John Beard noted that trophoblastic cells and tumor cells accomplish invasion of the surrounding tissue, grow, recruit a blood supply, and suppress the host immune system; the mechanisms used by both cells are the same [23].

### 2.1. The Evolutionary Theory of Carcinoma Evolution

In the past century, efforts to develop theories of the evolution of carcinoma have been proposed since the decade 1950, when authors such as Fisher and Hollomon (1951) and Nordling (1953), Armitage and Doll (1954) through the decade of the seventies proposed a multi-stage model of carcinoma progression. This theory could not be tested for its ability to account for discrepancies in driver mutations. Unfortunately, the multi-stage model is deficient in considering the fundamental evolutionary processes that shape the evolution of animal life history traits [28].

One of the bases of modern biology is that biological mechanisms cannot be understood unless linked to evolutionary processes [29].

Others have proposed that the oncogenic development could be equated to the social phenomenon of anarchy in that the “cellular anarchy” could destroy the cellular society as the accumulation of ideas and actions can collapse the governance of a society. Considering that a tumor could start within a single stem cell that mutates and begins to divide, producing cells that behave autonomously of the surrounding cells as the original cell but with independent mutations that give rise to the multiclonality within a given tumor [30].

With these tools, others have described the Darwinian basis of the oncogenic process; unfortunately, the role of events in tumor initiation contradicts the central Darwinian thesis, the gradualism; moreover, in some tumors exists what has been called neutral evolution; despite this, some have correlated the carcinoma evolution that is dependent on the aging of the genome [31], but multiple studies have shown that approximately 50% of mutations accumulated before maturity [28].

### 2.2. Carcinoma and Development

Some authors, such as Panje and Ceilley (1979), have discussed the link between embryonic development and the spread of head and neck skin carcinoma. They described how the embryology of the mid-face can influence the spread of epithelial malignancies [32]. Additionally, Höckel et al. [33] proposed the ontogenetic carcinoma field theory, which explains the oncologic process as a form of reverse morphogenesis driven by the reactivation of developmental programs controlling tissue morphogenesis for the development of the female genital tract. On this basis, this group of research defines mega compartments and the Müllerian compartment and from this subcompartment [34]. This ontogenetic theory is based on the pathological reactivation and maintenance of sequential developmental programs that control the morphogenesis of the tissues from which the carcinoma originated. These programs follow an inverse developmental sequence [33,35,36], suggesting that epigenetic processes are involved, independent of additional mutations, due to inherent qualities acquired during this developmental process [37].

Building on Pierce’s ideas, Cofre and Abdelhay support the hypothesis of carcinoma as an embryological phenomenon, similar to postembryonic differentiation. They also found that certain malignant tumors at the molecular level share a developmental signature with developing tissues, characterized by a set of genes activated simultaneously during embryogenesis, which they call the carcinoma transcriptome [38]. 

## 3. Molecular Embryology of Head and Neck

During embryo development, molecules can be grouped into transcription factors and signaling molecules that act as growth factors binding to receptors. In the craniofacial region, the embryonic players involved are the neural tube (that gives rise to the NCC), paraxial mesoderm, anterior visceral endoderm (AVE), representing the endoderm of the pharyngeal pouches, and the cranial ectoderm. This region develops in the fourth to fifth week and continues until the late teens [39,40].

The front of the head (primordia) is made up of the frontonasal prominence (unpaired), the nasal-medial and nasal-lateral processes (both paired), as well as the paired maxillary and mandibular processes. The frontonasal prominence and the forebrain are not as distinctly segmented as the pharyngeal region (pharyngeal arches) or the hindbrain [39,40].

In the tissue development of the craniofacial region, the AVE stimulates the facial ectoderm (ectodermic frontonasal zone). This, in turn, induces the neural crest mesenchyme precursors of the facial bones [39,40].

The AVE consists of specialized migratory extraembryonic epithelial cells that function as a signaling center and are responsible for multiple patterning events [41]. These cells are specified shortly before implantation and are derived from the distal visceral endoderm [42].

As the bilayer embryonic disk elongates, the visceral endoderm expands to cover the trophectoderm-derived extraembryonic ectoderm and the distally located epiblast. The asymmetric expression of *OTX2* and *DDK1* influences it [42,43]. A subset of cells from the distal visceral endoderm transforms into the AVE due to the interaction of *Nodal* and *MAPK* signaling pathways in response to *SMAD2* phosphorylation. This results in the differentiation of AVE, and these cells express specific markers such as *LEFTY I*, *CER I*, and *HEX* genes [41,44]. The regional organizing role of AVE could explain why the primary malignant tumors in the head and neck originate from the endoderm of the pharyngeal pouches (squamous cell carcinoma).

The development of the forebrain and its associated structures is influenced by genes such as *LHX1*, *EMX1*, *EMX2*, *OTX1*, and *OTX2*. These genes work with inductive signals on the prechordal mesoderm or the AVE [39,40]. Unlike the well-defined segments in the pharyngeal region and the hindbrain, the frontonasal prominence and forebrain are less segmented. Each arch in the frontonasal prominence and forebrain consists of a pouch (endodermal origin), the arch itself (mesenchymal origin), the grooves that divide the arches, and the ectodermal covering [39,40].

The first pharyngeal arch is unique because it does not rely on *HOX* genes for its development but instead depends on the action of *OTX2*. The development of the pharyngeal pouches depends on exposure to retinoic acid (RA). The second pouch requires only a tiny amount of exposure. In contrast, the rest of the pouches require a significant amount of RA in an increasing tissue gradient, except for the first pouch, which does not depend on the presence of RA [39,40].

The hindbrain segmentation (rhombomeres) is determined by the pattern of expression of *HOX* genes, which the NCC expresses. These cells invade the pharyngeal arches through three streams. The first stream is made up of NCCs derived from the first and second rhombomeres that invade the first arch, the second stream originates from the fourth rhombomere and ends in the second arch, and the third stream comes from the 6th and 7th rhombomeres and invades the rest of the arches [39,40].

### 3.1. Molecular Development of the Face (Pharyngeal Arches, Emphasize the First Arch)

During the early stages of facial development, the forebrain serves as a signaling center by expressing Shh. This stimulates the development of the frontonasal ectodermal zone from the ectoderm and produces RA. This zone is located at the tip of the nasomedial processes. It expresses FGF-8 and Shh to differentiate the NCC mesenchyme in the frontal-nasal process from the maxilla primordia [39,40].

The pharyngeal arches have a molecular organization that allows for polarization in the dorsoventral (proximal-ventral) and mediolateral (oral-aboral for the first arch) axes. The known genes involved include *ALX*, *HAND*, *MSX*, and *PRRX* [39,40].

The mesenchyme of the first arch is derived from the NCCs of the rhombomeres 1 and 2 of the hindbrain. While it was previously thought that the maxilla mesenchyme is derived from the rhombomeres 1 and 2, it is now known to contain a mixture of NCC originating from the forebrain and midbrain. However, the mesenchymal cells are derived exclusively from the rhombomeres 1 and 2 in the lower maxilla [39,40].

The specific morphology of the face skeleton is determined by signals from narrow endodermal areas that act on the ectoderm of the facial prominences via FGF-8. The face’s size and form depend on the WNTs that act on the tissues, stimulating cellular proliferation and, therefore, the organ’s size [39,40].

The first arch subdivision in the dorsal-ventral axis depends on the expression gradient of FGF-8, which is more concentrated in the distal region of the first arch, and bone morphogenetic proteins (BMPs), which are elevated in the proximal region. Endothelin-1 participates in the constitution of this dorsoventral axis. In the proximal region of this arch, low levels of endothelin-1 favor the expression of the distal gene pairs *DLX* 5 and 6, which induce the maxilla and the ear ossicles. In the distal region, high levels of endothelin-1 block proximal genes *DLX1/2* and stimulate distal genes *DLX5/6* and their downstream genes *HAND-2* (which inhibits the action of *DLX5/6*, acting probably as a stop signal) and *GOOSECOID*, permitting the patterning of the mandible. In the midportion of this arch, endothelin-1 through *DLX3/4* permits the expression of *BARX-1* for the temporomandibular joint patterning. Finally, in the medial-lateral axis, the expression and action of *LBX-1* determine the medial (oral) part of the mandible, and the action of *GOOSECOID* in the lateral (aboral) region determines the development of the mandible [39,40,45].

### 3.2. Some Strategic Genes in Head and Neck Development

The body structure in humans and vertebrates includes cells with four different embryonic origins: ectodermal derivatives, mesodermal derivatives, endodermal derivatives, and NCC and neural tube derivatives, named for some authors’ fourth germ layer [46]. The involved genes are part of representative marker genes for each of the above-mentioned germ lines; some of them are mentioned in the following paragraphs.

The *EMX* (empty spiracles homeobox) genes 1 and 2 belong to the *EMX* gene family of homeodomain genes [47]. They are categorized as transcription factors that regulate cell proliferation, migration, and differentiation during neural tube and neural crest migration. These genes are involved in specifying neural positional identity, promoting neural stem cell proliferation, and influencing the differentiation of different types of neurons during early embryonic and neural development [47].

#### 3.2.1. Wnt Signal

The Wnt signaling pathway is a critical factor in the developmental process of embryos, and it is conserved across different species. Mutations in various signaling pathways, including TGFβ, Hedgehog, and Wnt, have been strongly linked to developmental abnormalities in species ranging from Drosophila to humans [16,48].

The Wnt signaling pathway consists of Wnt ligands (such as Wnt3, Wnt4, and Wnt10a), Wnt receptors (Frizzled and LRP5/6), and downstream signaling molecules (including *GSK-3β*, β-catenin, and c-Myc). This pathway regulates cell proliferation, survival, and stem cell self-renewal. Wnt is crucial in signal transduction cascades during embryo development and maintaining tissue adult homeostasis [49]. In normal embryogenesis and adult tissue homeostasis, Wnts regulate cell motility, adhesion, invasion, tissue patterning, and proliferation [50,51].

Cell functions are closely connected to Wnt signals during development, childhood, and adulthood. Wnt genes contain a set of ligands that have been conserved throughout species evolution, and they trigger signaling pathways involved in cell proliferation, cell fate determination, apoptosis, and cell polarity during embryonic development. In adults, Wnt pathways are linked to maintaining stem cell populations [51,52,53].

The primary action mechanism in the Wnt pathway is called the “canonical” pathway, and it depends on the stabilization of β-catenin to activate transcription. When Wnt binds, its receptors create a blockage in the destruction complex. If Wnt is absent, this destruction complex causes β-catenin phosphorylation, leading to degradation mediated by the proteasome. As a result, β-catenin does not enter the cell nucleus, affecting transcription [16]. Mutations in the components of the pathway can cause significant developmental defects. In postnatal life, these mutations can also lead to or contribute to various carcinoma developments.

Currently, 19 ligands have been identified for Wnt, some functioning through canonical and non-canonical pathways [54,55]. The cellular mechanism modulated by Wnt ranges from stem cell self-renewal to cell motility, and β-catenin is a critical component in many Wnt pathways in normal and malignant transformed tissue.

We can establish links between aging-linked processes and Wnt pathways during aging, but it needs to be clarified [16]. Wnts can harm or benefit the aging process in worms, depending on the specific Wnt ligand, timing, and tissue involved. The suppression of Mom-2/Wnt is related to a shortened lifespan, whereas suppression of lin-44/Wnt can expand the lifespan [56].

The *OTX1* (orthodenticle homeobox protein one) and *OTX2* (orthodenticle homeobox protein two) genes play a crucial role in the embryonic development of the neuroectoderm. These genes encode transcription factors with homeobox-containing genes located in 2p13 and 14q21-22 on human chromosomes, respectively [57]. They are essential for the specification, regionalization, and late differentiation of the rostral structures of the central nervous system [58]. Additionally, they are vital for specifying cell identity, cell differentiation, positioning the midline (axis) of the embryo body [59], and other pleiotropic actions during embryo development.

During embryological development, *OTX1* is expressed in proliferative regions in the fetal brain’s neocortex, while *OTX2* expression is located in the diencephalon, mesencephalon, and archicortex [60].

In mice, double deletion of *OTX1* and *OTX2* is lethal. Following gastrulation damage, heterozygous double mutants are associated with central nervous system defects and abnormal sensory organ formation [61].

During early embryonic development, *OTX2* plays a critical role in maintaining ESCs in a state that allows them to fluctuate between different levels of pluripotency for differentiation and morphogenetic processes related to gastrulation [62]. In the embryo, *OTX* genes regulate the dedifferentiation in the epithelial–mesenchymal transition; in the adult, *OTX2* acts as a homeogene to prevent dedifferentiation by regulating target genes and preventing abnormal epithelial–mesenchymal transition (EMT) [62].

The *LIM* homeobox gene family plays a crucial role in organogenesis, cell-cell signaling, gene expression, transcription regulation, anatomical structure formation, cell differentiation, and growth. LIM proteins are essential for the development of almost all parts of the body. The absence of specific LIM proteins can lead to headless mammalian embryos [39,40].

Moreover, the *LHX1* and *OTX* gene families are associated with cellular activation and H&NSCC [21]. In addition, *LHX1* is closely linked to metastasis [63].

Bone morphogenetic proteins (*BMPs*) are part of the TGF-β superfamily of cytokines. Initially, they were found to promote bone formation during embryonic development, in contrast to their antagonistic counterparts, *NOGGIN* and *SOSTS* [64].

In vertebrates, early development progresses sequentially from anterior (front) to posterior (back). The earliest tissues are located at the front, and tissues at the back develop progressively later, in coordination with the temporal and spatial front-to-back patterning [65].

*BMP* signaling is a crucial pathway that regulates the development of the face and skull. It plays a key role in early head patterning, cranial NCC formation, and facial feature patterning. These proteins are essential for developing mineralized structures such as cranial bones, the maxilla, mandible, palate, and teeth. Genetic variations and mutations in *BMP* genes are associated with syndromic and non-syndromic craniofacial malformations. 

*BMPs* have a precise local regulation to control cell growth, cell death, interactions between different types of cells, and the differentiation of stem cells during the development of the face and skull [66]. Therefore, *BMP* signaling contributes to the shape and function of facial structures in humans. After birth, craniofacial growth is regulated, particularly concerning the lifelong development of dental structures and the growth, maintenance, and repair of craniofacial tissues.

These *BMP* antagonists induce a secondary axis and rescue A-P axes in ventralized embryos while regulating the spatial pattern of *HOX* gene expression. In conclusion, BMP establishes the trunk–tail axis and controls *HOX* gene expression [65,67,68].

Some lower vertebrate species, like frogs, may use a *BMP*/anti-*BMP* dependent mechanism to translate the time and location of *HOX* genes’ trunk and tail patterning. The *HOX* genes activate sequentially in a high mesoderm (non-organizer mesoderm) BMP region with dynamic and unstable expression [65,67,68].

In experiments on frogs, researchers investigated the role of *BMP* signaling in head patterning. They discovered a *BMP*-dependent timing mechanism that can ultimately result in spatial patterns through the involvement of anti-BMP signals [65]. This process is vital for the patterning of the vertebrate head.

To test the role of *BMP* signaling in head patterning, Zhu K et al. administered anti-*BMP* treatments at specific times to both wild-type (WT) and ventralized frog embryos. They observed that this resulted in either a halt in head patterning in WT embryos or a rescue of head patterning in ventralized embryos at different stages [65]. This suggests that a timing mechanism may be involved in patterning the vertebrate head, dependent on *BMP*, and can be converted into spatial patterns by anti-*BMP* signals [65].

*BMP* has regulative systems that affect extracellular, intracellular, and membrane functions. The extracellular regulators can either enhance or inhibit *BMP* signaling and include molecules such as Chordin, Crossveinless2, and Noggin from the CAN (Cerberus and DAN) family of proteins during gastrulation. Intracellular regulators of the *BMP* pathway include microRNAs (miRs), I-SMADS, and phosphatases, which interact with other signaling pathways such as Wnt [69].

Some other non-canonical, Smad-independent signaling pathways for *BMP* are part of a network. For instance, *BMP4* can activate *TAK-1*, a serine-threonine kinase of the *MAPKKK* family. Additionally, *BMP* is affected by PI3K/Akt, P/kc, Rho-GTPases, and others [70].

#### 3.2.2. *HOX* Genes

The homeobox genes consist of eight genes containing homeoboxes. The proteins in the homeodomain group share a highly conserved domain of 60 amino acids. The *HOX* genes, *PAX* genes, LIM proteins, and *DLX* gene family belong to this group. At least 39 homologous homeobox genes are arranged in 13 human paralogous groups. These genes are distributed in clusters in 4 mammalian chromosomes: *HOXA* (7p15), *HOXB* (17q21.12), *HOXC* (12q13), and *HOXD* (2q31). Homeobox genes regulate embryonic development and cell differentiation in eukaryotic cells [39,40,71,72].

Cells that are expressing *HOX* genes are receiving signals from the Spemann organizer. The *HOX* genes are located at different segments along the developing body axis, and the timing information carried by these genes determines the physical pattern of the body. It is believed that the signals from the organizer that help stabilize *HOX* genes are *BMP* antagonists, like Noggin and Chordin [65], which play a role in the dorsal development of the embryo [73].

The above scenario suggests that Noggin and Chordin in a developing embryo can rescue the head part of the axis, indicating the potential involvement of *BMP* signaling in head patterning. When *BMP* is blocked, the expressions of Six3 (a forebrain differentiation marker), Oxt2 (a forebrain and midbrain marker), Gbx1 (a rostral hindbrain marker), and Hoxb1b (a caudal hindbrain marker) are sequentially expanded. This raises the question: Is the *BMP* pathway involved in the progressive patterning of the head? [65].

#### 3.2.3. HIPPO Pathway

The pathway from flies to mammals is highly conserved. Its key components include *MST1/2-SAV1*, *LATS1/2* (and its scaffold protein MOB1), *YAP/TAZ*, and *TEAD 1-4*. This pathway acts as a cascade kinase with the primary objective of phosphorylating *YAP* and *TAZ* to prevent them from entering the nucleus. This is achieved through cytoplasmic retention or degradation of these proteins and by binding *TEAD 1-4* to activate gene expression. The cascade is regulated upstream and initiated with the phosphorylation of *MST1/2* and *MAP4Ks*; these then phosphorylate *LATS1/2*, which phosphorylates *YAP/TAZ* [24,74]. Dysregulation of the HIPPO pathway can cause various diseases, including eye, cardiac, lung, renal, hepatic diseases, and even carcinoma [74].

Although retinoic acid (RA) is not a gene, it is involved in the embryonic development of the head and neck. It is a fundamental component of vitamin A and is considered its active derivative (retinol). Its role is regulated through interactions with other developmental genes [45].

During embryo development, this molecule acts as a signaling morphogen essential for tissue differentiation. Dysregulation of its levels is associated with congenital structural alterations in the craniofacial structure and auditory and ocular abnormalities. This is due to disruptions in the cranial neural crest stem cells, a transient population contributing to the differentiation of various cell lineages during gastrulation and neurulation processes (see Table 1 and Figure 1).

RA plays a crucial role in regulating the development of cranial NCC during neural tube formation. These cells migrate to the craniofacial region, contributing to the formation of the face. RA regulates cranial NCCs, which form the frontonasal processes, periocular mesenchyme, and pharyngeal arches. Additionally, RA is involved in the development of bones, connective tissue in the head and neck regions, and the anterior segment of the eye through interactions between epithelial and mesenchymal cells. In adult life, RA also helps maintain the structures derived from NCC in adult stem cells (ASCs) (see Table 1) [45].

The development of NCC in the second to fourth pharyngeal arches is influenced by Hox (homeobox) transcription factors produced by cells before they migrate from the rhombomeres. In the lower vertebrae (i.e., frogs), Hoxa2 is activated at the boundary between the first and second rhombomeres, determining the characteristics of the second pharyngeal arch. Hoxa3 and Hoxa4 play roles in forming the third and fourth arches. Each pharyngeal arch’s development depends on a specific combination of Hox genes in the more posterior arches. RA is a crucial regulator of Hox genes within rhombomeres throughout this process, creating a gradient from the cranial to the caudal direction (see Table 1) [45,75].

## 4. Strategic Developmental Genes in Head and Neck Carcinoma

### 4.1. EMX1 and EMX2, Their Relationship with Wnt Regulation and Carcinogenesis

The *EMX1* and *EMX2* genes regulate the activity of stem cell regulatory genes (*OCT4*, *SOX2*, *KLF4*, *MYC*, *NANOG*, *NES*, and *PROM1*) in specific cancers. When the EMX proteins are downregulated, it can lead to malignant transformation, especially in sarcoma cases. It is believed that *EMX 1* and *2* may regulate the proliferation of neural crest-derived stem cells and act as tumor suppressors, negatively affecting carcinoma stem cells (CSCs) [47].

Several genes are categorized as transcription factors in certain solid tumors originating from epithelial and mesodermal tissues. Reestablishing normal *EMX 1* and *2* expression levels in these tumors has been shown to inhibit tumor cell growth and the tumor’s ability to spread in a treatment-sensitive phenotype [76]. In tumors with sarcomatoid transformation, high expression of *EMX1* and *EMX2* reveals [47] reverberant pathways that override the tumor suppressive role during the local progression of all tumor types.

Some authors suggest that downstream *EMX* protein transcriptional targets are linked to the direct inhibition of Wnt-1 through *EMX2*. This inhibition is achieved by the *EMX2* DNA-binding site located in a Wnt-1 gene enhancer [77,78,79]. The negative correlation between the *EMX* and Wnt pathways is predominantly observed in lung and gastric cancers.

### 4.2. Wnt Expression Is Related to Carcinoma

Wnt signaling is highly conserved across species, crucial for embryonic development, and implicated in breast and colon carcinogenesis [80]. 

If the Wnt signal is activated abnormally in tissues, it can lead to abnormal cellular behaviors, malignant transformation, and tumor progression, including metastasis of malignant cells [49]. The biological Wnt signaling pathway is involved in both the movement of cells and the invasion of surrounding tissue during both development and tumor progression. This process involves the ARF (ADP-ribosylation factor) and Rho-family small GTP-binding proteins related to regulating Wnt signal transduction. This regulation occurs from cell surface receptors, signaling endosomes, and extracellular vesicles, all tied to cell movement, extracellular matrix degradation, and paracrine signaling [49].

Dysregulation of the Wnt signaling pathway is closely associated with the loss of normal cell polarity and adhesion properties, which leads to motility and invasiveness during carcinoma progression and metastasis [81,82].

When the Wnt signaling pathway undergoes abnormal changes, it often leads to modified cellular behaviors that can initiate or exacerbate certain benign and malignant diseases (see Table 2). Signaling endosomes and extracellular vesicles facilitate these changes, which can impact the invasion of tumor cells through the degradation of the extracellular matrix [81,82].

Various processes, such as differentiation, adhesion, cell morphology, and motility, play crucial roles in normal embryo development. However, when these pathways are disrupted, they can promote carcinoma development, progression, and metastasis in adulthood. It is important to note that Wnts do not always contribute to tumor progression; their effects may be oncogenic or tumor-suppressive, depending on the specific microenvironment and cellular conditions [49].

### 4.3. Wnt in Head and Neck Carcinoma

In squamous cell carcinoma of the head and neck region, the progression and spread of the carcinoma impact treatment decisions. Certain epigenetic factors, such as *KDM5B*, can act as oncogenes, affecting patient prognosis. *KDM5B* overexpression is linked to poor disease-free outcomes and OS. Its impact is associated with cell migration and invasion processes, which are mediated by alterations in the Wnt/β-catenin pathway and epithelial-mesenchymal transition. Yuan W and other authors have demonstrated how inhibiting the Wnt/β-catenin pathway using a small molecule inhibitor called iCRT-14 can partially reduce the enhanced migratory and invasive ability caused by *KDM5B* in squamous cell carcinoma of the head and neck cells. This suggests that *KDM5B* promotes metastasis through the Wnt/β-catenin pathway and could serve as a therapeutic target and prognostic marker in H&NSCC [83].

Adenoid cystic carcinoma is a common malignant tumor found in the salivary glands. It is composed of myoepithelial and epithelial glands. During the progression and metastasis of these tumors, there is enhanced expression of Ki67. Additionally, there is high cell membrane expression of C-kit and phosphorylated *ERK*, which suppresses the β-catenin pathway and KIT expression. These results suggest that the inactivation of the Wnt/β-catenin signaling pathway may trigger C-kit-*ERK* signaling and cell proliferation, leading to metastasis in adenoid cystic carcinoma [84].

### 4.4. OTX1 and OTX2 in Head and Neck Carcinoma Oncogenesis

The expression of *OTX1* and *OTX2* genes has been recently reported in both non-transformed and transformed tissues, including the retina, breast, hypophysis, sinonasal mucosa, some cancers, and as part of the tissue response to inflammatory, ischemic, and hypoxic phenomena [61]. The *OTX* genes can reactivate in adult and differentiated tissues as part of the evolutionary amplification of gene function. The temporal expression of these genes can vary, and homeobox genes can be selected from the toolbox to aid, drive, or contribute to different mechanisms throughout adult life. The *OTX* genes have the potential to serve as diagnostic or prognostic markers and even as therapeutic targets (see Table 2).

**Table 2 ijms-25-09979-t002:** Carcinoma by anatomical location and involved genes and their mechanisms.

Carcinoma	Gene	Mechanism	Observations	Reference *
Nasopharynx	*Notch 3*	Enhanced expression	Inhibition is related to high response to cisplatin in presence of EBV	[85]
	*SLUG*	Enhanced expression	Radio resistance	
	*IGF2-BP3*	Associated to M6A	Constitutive activation Notch 3	[85]
	*LHX2*	Enhanced expression	Poor prognosis	[86]
	*FOXA1*	Enhanced expression	Good prognosis associated to EBV	[87]
	*PAX5*	Diminished expression	Advanced clinical stage at diagnosis and poor prognosis	[88]
Sinonasal	*TP53*	Mutated	Poor prognosis	[89]
	*EGFR*	Amplification, increased number of copies, enhanced expression, activator mutation in exons 19, and 20	Poor prognosis	[89]
	*P16*	Mutation associated to HPV-16, 18, 31 and 33	Improve overall survival	[89]
Oral squamous carcinoma	*SHH*	Abnormal activation	Promoting invasion and metastasis	[89,90]
	*HOXA2*	Upregulated	Oral dysplasia	[43]
	*HOXA7*	-	Less aggressive phenotype	[43]
	*HOXA10*	-	Less aggressive phenotype	[43]
	*HOXA13*	Overexpression	Good prognosis	[44]
	*HOXB2*	-	Present in malignant lesions	[43]
	*HOXB7*	-	Pivotal role in metastasis	[43]
	*HOXC6*	-	Pivotal role in metastasis	[43]
	*HOXC10*	-	Pivotal role in metastasis	[43]
	*HOXD10-11*	Upregulated	Malignant phenotype	[43]
	*HOXD13*	Overexpression	Poor prognosis	[44]
	*WNT*	Abnormal activation	Drug resistance	[91,92,93,94]
	*BMP2*	Aberrant expression	Development and progression	[95]
Oropharynx	*HPV-16*	-	Improve overall survival	[96]
Larynx	*HPV-16* and *18*	-	Improve overall survival	[97,98]
	*OTX1*	Overexpression	Nodal metastasis, poor prognosis and low rates overall survival	[99]
	*HIPPO^+^*	Overexpression	Carcinoma development Poor prognosis	[100,101]

* See the complete reference at the end of the article.

The *OTX* genes are closely associated with the sinonasal mucosa’s acute and chronic inflammatory processes. They are expressed in lesions in the ethmoidal sinonasal tissues extending into the nasal cavity and middle turbinate [102]. The affected cells concurrently express p63 and its isoforms, TAp63 and delta-N-p63 [35], which are associated with polyps and have a pro-proliferative and oncogenic effect on the cells [103,104]. In this context, *OTX2* plays a crucial role in determining the progression and recurrence of the condition due to its interaction with TAp63. *OTX2* can transactivate TAp63, causing the cells to differentiate and enter a non-proliferative state, leading to a lower recurrence rate in cases where TAp63 is not expressed or has low expression.

The expression of the *OTX* gene plays a significant role in modulating neoplastic tissue in sinonasal neoplasms. This suggests that the activation or inactivation of *OTX* genes is associated with various sinonasal neoplasms. Molecular analysis using real-time PCR and pathological observations through immunohistochemistry have shown that *OTX1* and *OTX2* are expressed in non-transformed sinonasal mucosae. In contrast, *OTX2* is expressed in olfactory neuroblastomas and poorly differentiated neuroendocrine carcinomas. This finding indicates that *OTX* genes could potentially serve as markers for tumor prognosis and as targets for therapy [61,85].

*OTX* genes are expressed during embryo morphogenesis, and they continue to develop in the olfactory epithelium in adults and can cause mutations in adult human tissues, leading to malignant tumorigenesis. Micheloni G. et al. recently suggested that *OTX1* and *OTX2* could be considered molecular markers in developing nasal tumors. This was based on the protein expression in tissue (by immunohistochemistry) and the presence of the mRNA using real-time PCR. They identified the protein expression and the mRNA of *OTX2* in olfactory neuroblastomas. They suggested these molecules could be helpful as molecular markers for differential diagnosis in anaplastic lesions of sinonasal tumors [85,105].

### 4.5. BMP and Squamous Cell Carcinoma

*BMP* plays a central role in regulating the survival and maintenance of CSCs, contributing to disease recurrence and treatment failure in various malignancies, such as H&NSCC. During cell development, some of its intracellular signaling is regulated by *SMAD*-specific E3 ubiquitin protein ligase 1 (*SMURF1*). Khammanivong et al. reported that the expression of this protein in CSC increases *pSMAD1/5/8*, a marker of *BMP* pathway reactivation. Decreased *SMURF1* expression promoted adipogenic differentiation, indicating a loss of tumorigenic ability [106]. This suggests that the molecule could be a new therapeutic approach to promote differentiation and reduce CSC populations, potentially leading to reduced drug resistance and disease recurrence.

Cetuximab is a commonly used targeted therapy for oral and other squamous cell carcinoma. However, some patients have shown limited response or developed drug resistance. In these cases, the resistance is not due to changes in the epidermal growth factor receptor (EGFR) but instead to *BMP7*-p-Smad1/5/8 signaling. To address this issue, some researchers suggest combining cetuximab with a BMP signaling inhibitor as a potential new treatment approach for patients with cetuximab resistance [86].

### 4.6. HOX Genes during Craniofacial Development and Carcinoma Development

Research by Shenoy et al [20]. has shown that the abnormal expression of the homeobox gene (*HOX*) is involved in developing certain types of carcinoma. However, the regulation of the *HOX* cluster still needs to be fully understood. The researchers used comprehensive databases to identify the differentially expressed *HOX* genes based on clinical stage and HPV status [20,87]. They found that genetic, epigenetic, and post-translational modifications of essential genes in tumor progression are linked to *HOX*. This research can potentially improve carcinoma treatments associated with gene alterations in the HOX gene family in H&NSCC (see Table 2, Figure 1).

*HOX* genes can be expressed in an aberrant form, such as protein-coding genes and non-coding RNAs (ncRNA). These characteristics can be biomarkers for early carcinoma detection in high-risk populations with H&NSCC [88,89]. 

The developmental *HOX* genes play a specific role in the growth and differentiation of segmental body regions [107]. The 39 *HOX* genes are grouped in four clusters, and abnormal *HOX* gene expression is implicated in developmental anomalies and malignant cell transformation [108,109,110]. Epigenetic factors regulate these expression changes, including promoter DNA methylation and ncRNAs [90,111,112].

Most *HOX* genes have unique ncRNA at their 3’ untranslated regions. The *HOX* loci contain 231 *HOX* cluster-embedded ncRNAs with broad implications for development and disease. The interaction between each HOX cluster member and its downstream biological target is essential for the proper functioning of organs at the tissue and cellular levels [113,114]. All of them are involved in abnormal proliferation, migration, invasion metastasis, and altered EMT during carcinogenesis [72,115].

In a stem cell setting, these cells are undifferentiated and have the unique ability to renew and develop into different types of cells [115]. The core gene network regulates their functions. *HOX* genes are crucial in regulating embryo development and differentiating specific lineage cells. If these genes are abnormally expressed in adults, it can lead to the development of malignant tumors.

The *HOX* genes are genes passed down through evolution and are active during the development of embryos in vertebrates [115]. In humans, these genes are divided into groups (A, B, C, and D), and there are 39 genes. They become active in the caudal portion of the primitive streak of the embryo. The order of the HOX genes within each cluster reflects their activity level along the front-to-back axis, with expression occurring from the 3’ (front) to the 5’ (back) direction. This is known as posterior dominance. The expression of these genes is influenced by Wnt ligands, RA, and fibroblast growth factor (FGF), which form a signaling pathway to control *HOX* gene expression [115].

The expression of the *HOX* gene is related to the organization of chromosomal clusters, cis-regulatory elements (enhancers and long non-coding RNA), histone modifications, chromosome boundaries, and 3D chromatin conformation in a way that follows the order of the genes along the chromosome. During embryo development, the regulation of HOX is achieved by the methylation of Histone H3 residues by the trithorax (TrxG) and Polycomb (PcG) group of proteins. These epigenetic modifiers create bivalent domains in the pre-gastrulation embryo by controlling the expression of developmental genes through transcriptional mechanisms. Along with miRNA and long non-coding RNAs, they regulate the expression of *HOX* genes during embryo patterning and morphogenesis [115].

In carcinoma, alteration in *HOX* gene expression plays a critical role as they are frequently deregulated. Upregulation of *HOX* genes, generally expressed in undifferentiated cells, drives oncogenesis, while their usual expression acts as a tumor suppressor [91,115].

In H&NSCC, the expression of *HOX* homeobox is dysregulated, especially the protein-coding *HOX* genes *HOXA10*, *HOXC9*, *HOXC10*, and *HOXC13*, during the development and progression of this type of carcinoma. Additionally, 31 out of 39 mammalian *HOX* genes were upregulated in H&NSCC. Moreover, in H&NSCC, *HOXC9*, *HOXC10*, and *HOXC13* genes are involved in cell migration, proliferation, and cell cycle progression. This suggests they play a role in the carcinoma’s transformation by influencing tumor growth and metastasis [72,92].

Essential changes in *HOX* genes were linked to *TP53*, *FAT1*, and *CDKN2A* mutations in H&NSCC patients. The interactions between these factors disrupt epithelial/mesenchymal transition, cell cycle, and apoptosis in H&NSCC progression [87].

### 4.7. RA and Carcinoma Development

In H&NSCC, retinoids have been suggested as a way to prevent a second primary tumor in the head and neck region or to reduce the chances of recurrence. Retinoids work by binding to and activating specific RAid receptors [93]. These receptors move into the nuclei of cells, forming homo and heterodimers, which then bind to specific DNA sequences in the regulatory regions of target genes. They also recruit transcription co-activators crucial for cell growth, differentiation, survival, and malignant transformation. RAid receptors can suppress gene expression, countering the activity of other nuclear transcription factors. In addition, RA induces growth arrest and cellular differentiation, thereby improving the effectiveness of chemotherapy and radiation treatment for H&NSCC [93].

The Notch signaling pathway is evolutionarily conserved. The core components of *NOTCH* signaling are highly conserved from Caenorhabditis elegans to mammals, and this pathway operates in most multicellular organisms [94]. It plays crucial roles in regulating development and homeostasis in various tissues, including lineage commitment, cell cycle progression, differentiation, and maintenance of self-renewal capabilities in stem cells. The signaling pathway plays a crucial role in developing various cancers by regulating stem and progenitor cells within the same tissue at different times. This pathway functions through five ligands of the *DSL* family and four receptors (*NOTCH 1-4*) [95].

In this context, it is a complex process of reverberation involved in various squamous cell cancers and exhibits pleiotropic behavior. At times, it demonstrates affluent oncogenic characteristics, while in other instances, it displays anti-oncogenic traits. The role that *NOTCH* plays in the development and progression of squamous cell carcinomas is associated with the loss of *NOTCH* signaling, and this is likely to be an early event in carcinogenesis [95]. 

At least in 50% of acute lymphoblastic T-cell leukemia, there is a mutation in the extracellular NRR (negative regulatory region) that ensures the *NOTCH* signaling is inactive in the absence of interaction with specific *NOTCH* ligands. A second mutation reported in *NOTCH* is in the *PEST* domain that is typically phosphorylated by *CDK8* and converts this *PEST* region into a phosphodegron that is ubiquitylated rapidly by a ubiquitin ligase (FBXW7 E3), and this induces the proteasome-mediated degradation of the intracellular domain of *NOTCH* [94]. In this manner, Notch, in its oncogenic capacity, promotes growth, invasion, and EMT and predisposes to carcinoma stemness and chemoresistance [100,116]. 

## 5. Head and Neck Carcinoma by Anatomical Regions 

### 5.1. Nasopharynx (For This Section, See Figure 1 and Table 2)

Nasopharyngeal carcinoma (NPC) is a rare form of carcinoma, comprising 0.7% of all cancers diagnosed globally in 2018. However, it is considered endemic in Southern Asia, Eastern Asia, North Africa, Micronesia, and Polynesia. In these areas, there is a high likelihood of the carcinoma spreading to distant parts of the body, impacting about one-third of patients in the highest-risk groups. The carcinoma is linked to Epstein–Barr virus infection, which is an etiologic factor, and it has a worse prognosis compared to NPC associated with HPV [96,97]. NPC is related to embryonic genes such as *NOTCH*, *FOXA-1*, and *LHX-2* (see Figure 1 and Table 2).

The *NOTCH* signaling pathway has been highly conserved throughout evolution. In mammals, it consists of four receptors (*NOTCH 1-4*) and five ligands (*DLL1*, *DLL3*, *DLL4*, *JAG 1*, and *JAG 2*). This pathway is involved in developing and maintaining multiple organs; its changes are linked to both cancerous and noncancerous diseases [98].

During development, the *NOTCH* pathway plays various roles as an inhibitory intranuclear signal in the lateral inhibition phenomenon (LIP). Cells expressing the delta receptor inhibit the same lineage differentiation in neighboring cells [40]. In various types of carcinomas, the *NOTCH* signaling pathway is overactive. This pathway controls the stemness and metastasis of carcinoma cells [117].

In NPC, *NOTCH-1* and *NOTCH-3* exhibit higher expression levels than normal tissues. Inhibition of *NOTCH-3* is linked to increased sensitivity to cisplatin treatment in NPC associated with EBV infection. Liu et al. reported that enhancer remodeling activates *NOTCH-3* in an aberrant way, conferring chemoresistance to these cells through the enhanced expression of *SLUG*, a member of the *SNAIL* family. *SLUG* regulates EMT and also plays a role in maintaining stemness properties while being associated with radioresistance [118]. 

As previously mentioned, NPC has a high potential to spread to other body parts. Studies have shown that inhibiting the *NOTCH* signaling pathway reduces NPC cell proliferation and increases their sensitivity to treatment. M6-methyladenosine (m6A) is a post-transcriptional modification associated with multiple cancers. The abundance of m6A is critical for carcinoma initiation, progression, metastasis, relapse, and treatment resistance. NPC can exert high metastatic activity through the m6A readers that prevent the decay of m6A-modified mRNA; one of these readers is the insulin-like growth factor 2 mRNA-binding proteins 1–3 (*IGF2BP1-3*) [117]. Chen et al. showed that IGF2BP3 binds to m6A-modified *NOTCH-3* and enhances mRNA stability, leading to continuous activation of *NOTCH-3* signaling [117].

Additionally, Xie et al. demonstrated that *LHX2* (LIM-homeodomain 2) is a transcription factor belonging to the LIM protein family, is involved in developing various body parts, and is upregulated [119]. The overexpression of *LHX2* is associated with poor survival due to increased growth, migration, and invasion in both in vitro and in vivo settings. This is linked to elevated transcription and FGF-1 expression, which activates the phosphorylation of *STAT3*, *ERK 1/2*, and the AKT signaling pathways [119].

The Forkhead box (*FOXA*) is a family of transcription factors (*FOXA1*, *FOXA2*, *FOXA3*) that regulate gene expression and chromatin structure. They are expressed in developing organs such as the liver, pancreas, lung, and mammary gland, and they have distinct domains that work together to direct morphogenesis [99]. In breast carcinoma, overexpression of *FOXA1* is a good prognosis marker in ER+ breast carcinoma, but in prostate, epithelial ovarian, and gastric cancers, it indicates poor prognosis [99]. Ammous-Boukhris et al. studied 56 NPC patients and found that *FOXA1* expression is present in 60.7% of patients, and this is associated with the TNM stage and age at diagnosis (low TNM and older age at diagnosis). In this study, low age at diagnosis is more frequent in the presence of LMP-1 (EBV oncoprotein), and high levels of *FOXA1* are associated with nonaggressive behavior and good prognosis in NPC [99].

In a study of NPC, Ye J. and colleagues found that a potential tumor suppressor molecule, *PAX5*, had decreased expression in malignant tissues. This decrease was associated with advanced clinical stage and poor prognosis. The researchers also suggested that PAX5 may be involved in various pathways related to cell cycle and tumor processes by dysregulating the Wnt signaling pathway, which activates the transcription of multiple oncogenes [120].

### 5.2. Sinonasal Carcinoma

Sinonasal carcinomas, which occur in the nasal cavity, frontal, sphenoid, ethmoid, and maxillary sinuses, are rare and comprise 3–5% of all head and neck cancers. These cancers are known for their aggressive nature and often consist of high-grade or undifferentiated cells. Molecular tests in clinical settings can help identify unique subsets of sinonasal carcinomas. Those with squamous histology can be further classified based on their association with HPV infection, *EGFR* alterations, or chromosomal translocations such as *DEK::AFF2*, *ETV6::NTKR*, etc.

Squamous cell carcinoma is the most common histotype, accounting for 60% to 75% of cases. Most cases feature mutations in TP53, especially the keratinizing subtype, with mutation frequencies ranging from 33% to 100%. This mutation is often linked to wood dust exposure and a poor prognosis. The second most frequent alteration is in the EGFR gene, which may manifest as amplifications, copy number gains, protein overexpression (in approximately 40% of cases), or activating mutations (in 6% to 15% of cases, mainly in exons 20 and 19).

The role of HPV infection in sinonasal carcinoma is still debated. The presence of P16 alone or with other HPV species ranges from 11.4% to 31.1%, and commonly associated HPV genotypes include HPV-16, 18, 31, and 33. As with oropharyngeal carcinoma, the presence of p16 is linked to improved survival. The clinical and molecular significance of squamous sinonasal carcinomas arising from sinonasal papilloma is an ongoing debate and interest [121].

### 5.3. Oral Carcinoma

Cancers of the oral cavity, including the oral mucosa, upper and lower alveolar ridge, retromolar trigone, floor of the mouth, hard palate, and anterior two-thirds of the tongue, collectively rank as the 16th most common malignant neoplasm globally. Almost 630,000 new cases are reported yearly, with a five-year survival rate of only 50% [101]. Squamous cell carcinomas account for over 90% of oral cancers, and around two-thirds of cases are documented in developing countries. The risk of developing oral carcinoma rises with age, and most cases are diagnosed in individuals over the age of 50 years [97,122,123].

In oral squamous cell carcinoma (OSCC), the overexpression of various genes, such as *OCT4*, *SOX2*, *NANOG*, *HOX*, and *SHH*, has been observed. *OCT4*, *SOX2*, and *NANOG* are embryonic transcription factors expressed by a CSC group expressing CD44. The *NANOG* signaling axis stimulates the overexpression of miR-21, which is associated with a poor prognosis in laryngeal cancers [124].

The *Sonic Hedgehog* (*SHH*) signaling pathway is a fundamental stem cell pathway. It plays a crucial role in embryo development and is quiescent in adult tissues. However, when the SHH pathway is abnormally activated, it can contribute to the malignant transformation of head and neck tissues, such as in oral squamous cancers [125,126].

Research indicates that Shh signaling plays a role in the development and advancement of different cancers, including oral carcinoma, through two molecular mechanisms: (a) the classical or canonical pathway, which involves the binding of Shh ligands and the activation of the *GLI* (glioma-associated oncogene) family, and (b) the non-canonical pathway, which involves cross-talk or downstream activation of one of the pathway members [125].

The Shh signaling pathway molecules expression in OSCC may be a prognostic marker, and it can stimulate the angiogenesis of tumor-derived vascular endothelial cells through paracrine mechanisms in the tumor invasive boundary [125], promoting invasion and metastasis. In the tumor microenvironment, Shh has an autocrine effect initiating cell proliferation and paracrine with the anomalous parenchyma–stromal interaction in OSCC through altered epithelial–mesenchymal transition, involving *BMP* and *BMP* receptor families [125,127]; both joined to another redundant mechanism such as Notum potential pro-survival circuit through cross-talking between Shh and Wnt/β-catenin signaling using the via phosphorylation of glycogen synthase kinase-3β [128,129].

Homeobox genes (*HOX* genes) normally regulate embryonic development and cell differentiation in eukaryotic cells. It has a crucial role in neoplastic transformation, specifically the group of the *HOX* 13 genes (*HOXA13* to *HOXD13*) in different cancers (bladder, pancreatic, liver, thyroid, metastatic melanoma). In OSCC, the overexpression of HOXA13 is associated with a good prognosis, but the overexpression of *HOXD13* is associated with a bad prognosis [72]. This observation is reinforced by the findings of Padam KS et al. describing the *HOXA2* upregulation in oral dysplasia and the loss of *HOXB2* expression in malignant oral lesions. *HOXA7*, *HOXA10* (regulates proliferation, migration, and invasion and is associated with a less aggressive tumor phenotype), *HOXB7*, *HOXC6*, *HOXC10* (regulates oral tumorigenesis through *Wnt-EMT* signaling pathways and may play a pivotal role in metastasis in OSCC), *HOXD10*, and *HOXD11* were upregulated in oral carcinoma; in patients with an added risk factor, *HOXA10* was involved in transcriptional misregulation, contributing to the malignant phenotype in oral carcinoma [71].

In OSCC, the presence of molecules such as *CCT6A* (Chaperonin containing TCP1 subunit 6A) is involved in carcinoma pathogenesis and progression through interrelated pathways. Chen et al. observed that proliferation, apoptosis, and invasion are related to the overexpression of Wnt14 and Notch1, revealing how Wnt and Notch’s pathways are involved in the processes through the effect of *CCT6A*, which is active in both [130].

In the pathogenesis of OSCC, members of the Wnt pathway family activate the TCF receptor gene and prevent the degradation of β-catenin. This leads to the transactivation of target oncogenes, resulting in cell growth and invasion. On the other hand, abnormal Wnt pathway activation also involves inhibiting chemotherapy-induced apoptosis, which triggers drug resistance [131,132,133,134]. In the cancers of this anatomic region, alterations of BMP expression and aberrant signaling pathways are associated with the development and progression of OSCC, and *BMP 2* mediates those alterations [135].

### 5.4. Oropharynx and Larynx Carcinoma

Due to the association between oropharyngeal and laryngeal carcinoma with HPV and the already demonstrated role of HPV in cervical carcinoma, we describe here both cancers and the differences between them in their carcinogenic process.

Oropharyngeal squamous cell carcinoma (OPSCC) includes tonsils, soft palate, uvula, and tongue base cancers. It is linked to alcohol and tobacco consumption, but the incidence associated with tobacco use has been decreasing in high-income countries. Concurrently, the occurrence of (HPV+) OPSCC has been increasing, especially in young people globally [96]. The global incidence and mortality rates are expected to increase by approximately 50% over the next 20 years [136].

Laryngeal carcinoma is the second most common type of H&NSCC [137]. The ratio of men to women affected is 8:1 [124]. The prevalence in the larynx is divided into three regions: supraglottis (30% to 35% of cases), glottis (60% to 65% of cases), and subglottis (5% of cases). In 2016, there were an estimated 13,430 new cases of laryngeal carcinoma, resulting in approximately 3620 deaths. Several risk factors have been linked to the development of laryngeal carcinoma, with tobacco and alcohol being the most significant. Smoking can increase the risk 10 to 15 times and even up to 30 times in heavy smokers. At the same time, there is a linear relationship between the amount of alcohol consumed and the risk of developing laryngeal carcinoma [97,138].

In oropharyngeal carcinoma, the most common subtype associated is HPV-16. Studies have reported HPV-16 with prevalence rates of 50–60%, and some authors suggest that the prevalence of HPV-16 can be as high as 80–90% [139].

In laryngeal carcinoma, HPV subtypes associated with the disease are mainly HPV-16 and HPV-18, with a higher proportion of HPV-16, which have been detected in up to 33% of cases. Similarly, oropharyngeal carcinoma can result from a non-transforming HPV infection [138,140].

The prevalence of OPSCC linked to HPV (HPV+) varies among countries, ranging from 0% in southern India to 85% in Lebanon. Among (HPV+) cases, 85% to 96% are associated with HPV-16. Globally, the incidence of (HPV+) OPSCC was reported to be 33% in 2021. However, other studies have shown a prevalence of HPV DNA of 23.5% and 24% for OSCC and laryngeal squamous cell carcinoma, respectively [139,141,142].

The association between (HPV+) and (HPV−) OPSCC has important clinical implications, leading to separate treatment approaches for each. Despite standard aggressive therapy, less than 25% of patients will still experience a return of carcinoma in the same area (locoregional recurrence) or the spread of carcinoma to other parts of the body (metastasis) within two years of treatment [139].

Due to this, Ang et al. categorized OPSCC into risk groups: those with (HPV+) and no history of tobacco exposure are considered low risk. In contrast, the intermediate-risk group consists of patients with (HPV+) and a history of tobacco exposure. Patients with (HPV−) fall into the high-risk group [139].

The current classification needs to fully explain the various clinical and morphological profiles. This suggests that there is genetic diversity. Gene expression analysis has identified three main molecular subtypes of (HPV+) OPSCC. The first subtype has low HPV integration, an enriched immune phenotype, and high mesenchymal differentiation. The second subtype has a high degree of keratinization and is basal like with high stromal content. The third subtype has a high keratinized profile, low stromal content, and suppressed immune responses.

Additionally, intratumoral heterogeneity, where different cell populations coexist, is present. Poorer survival is associated with aggressive or persistent cell populations. RNA sequencing has allowed for the classification of (HPV+) OPSCC into a KRT molecular subtype, which is characterized by an increased keratinization signature (worse prognosis and higher HPV integration), and an IMU subtype, which exhibits a more robust immune response and mesenchymal signature [139].

In oropharyngeal carcinoma, the highest prevalence is in the tonsils (56–64%), followed by the base of the tongue (40–56%). The HPV-attributable fraction was reported as 22% for OPSCC and 47% for tonsillar carcinoma [143]. The molecular basis of (HPV+) OPSCC, as in other malignancies associated with HPV, is similar, with the E6 and E7 viral proteins (oncogenes) being the key drivers. The greater the Rb and P53 dysfunction is, the greater the cumulative genomic instability associated with E6 and E7 [96]. Similarly, aberrant expression of p16, pRb, cyclinD1, and p53 has been reported in laryngeal carcinoma [140].

*OTX1* plays a role in the development and progression of laryngeal squamous cell carcinoma [61,144]. According to Tu XP. et al. [144], over-expression of *OTX1* is associated with higher rates of nodal metastasis, poor prognosis, and lower overall survival rates. They also discovered that *OTX1* is negatively regulated by miR-129-5p, which is closely linked to lymph node metastasis in laryngeal cancers. They suggested that increasing miR-129-5p levels could be a potential therapeutic strategy for patients with laryngeal squamous cell carcinoma and high *OTX1* expression [144].

MiRs are crucial in diagnosing and prognosing laryngeal carcinoma. MiRs are a class of non-coding RNAs that are small evolutionary molecules consisting of single-strand RNA of 22 nucleotides long, and they modulate gene expression by inhibiting mRNA translation or inducing mRNA degradation. They have critical regulatory roles in various biological processes in eukaryotic organisms [124,145]. MiRs regulate post-transcriptional gene expression by forming RNA-induced silencing complexes. In laryngeal carcinoma, certain miRs are associated with prognosis. For instance, the downregulation of miR-101 and miR-34c-5p and the upregulation of miR-19a, miR-23a, and miR-296-5p are linked to a poor prognosis.

On the other hand, the upregulation of miR-126 is associated with a favorable prognosis [124,146]. The *MAF* transcription factor also acts as a tumor suppressor in laryngeal carcinoma by regulating apoptosis. However, its function is lost in squamous laryngeal carcinoma due to the suppressing role of miR-1290 over *MAF* [124,146].

The HIPPO signaling pathway components (YAP and TAZ) have increased expression in laryngeal and other cancers. The HIPPO pathway is a crucial signal transduction pathway that regulates organ size, tissue repair, homeostasis, differentiation, and immunity. Disruption of this pathway leads to cell proliferation, migration, survival, and carcinoma development. This pathway can interact with other carcinoma-associated protein networks, such as *WNT*, *TGFb-BMP*, Hedgehog (*HH*), Notch, and *mTOR* at the *YAP* and *TAZ* levels [137,147]. High expression of HIPPO signaling pathway components, including *YAP*, *TAZ*, *TEAD4*, and *p73*, significantly correlates with high grade, tumor stage, supraglottic location, and recurrence, making them useful prognostic markers [137].

## 6. Head and Neck Squamous Cell Carcinoma and the Role of HPV in Oncogenesis

The connection between HPV and various types of head and neck carcinoma has been well established for over 40 years. HPV is linked with cancers in areas such as the oropharynx, mouth, larynx, and sinonasal. The significance of HPV in non-oropharyngeal H&NSCC has been overestimated due to the detection method used. Often used for detection, PCR can pick up even small traces of HPV–DNA, including potential contamination in the laboratory. Therefore, it is more reliable to detect transcripts of E6 and E7 to accurately determine the role of HPV in causing carcinoma [143].

Even though the role of HPV in H&NSCC has led to a better understanding of the development and outlook of this group of cancers, the majority of (HPV–) cases have a range of mutations in genes that suppress tumors and genes that can cause carcinoma [143,148]. According to the Carcinoma Genome Atlas, exome sequencing analyses have shown that mutations or deletions in the p53 gene have been found in 63% of cases [149] and 84% [63]. 

In this subgroup of cancers, 58% of (HPV−) H&NSCC show inactivating mutations in *CDKN2A*, a cell cycle regulator. Losses of function have also been found in the *TGFβR/SMAD* signaling pathways, including *TGFβR2*, *SMAD 2*, and *SMAD 4*. Studies on cutaneous squamous cell carcinoma suggest that *TGFβ* may have a dual role in oncogenesis, acting as a tumor suppressor in the early stages. However, once the process is ongoing, *SMAD4* promotes EMT and supports metastasis, especially when cooperating with *KRAS*. Additionally, 11–19% of tumors have documented loss of function in *NOTCH1*, *NOTCH 2*, and *3*, while *NOTCH1* can also interact with other genes associated with differentiation, such as IRF6 and TP63 [148].

Additional mutations have been observed in tumor-suppressor genes, including *FAT1*, which usually inhibits cell proliferation by preventing the nuclear localization of nuclear β-catenin. Mutations have also been detected in apoptosis-related genes (*CASP8, DDX3X*), histone methyltransferases (*PRDM9*, *EZH2*, *NSD1*), and Ajuba, a centrosomal protein that regulates cell division in a manner dependent on *EGFR-RAS-MAPK* [148].

The process of developing carcinoma in individuals with (HPV+) genotype 16 typically occurs after a persistent infection lasting 10–30 years, often in association with immunosuppression. The infection usually peaks at ages 30–34 and 60–64 [136]. Viral DNA integration within the host cell DNA, in both basal and mitotically active cells, has been observed in 74% of patients who test positive for HPV. At the same time, the remaining cases could lead to carcinoma through different mechanisms related to episomal HPV. The immune microenvironment of the tonsillar crypts plays a role in clearing the HPV infection, with HPV accounting for 3.9% of cases.

Additionally, resident myeloid cells may aid HPV in evading local immune surveillance, increasing the proportion of infections that transform, with HPV accounting for 47% of cases [143]. This phenomenon has been described by Hanahan et al. as “carcinogenesis corruption-induced”. The clinical significance of viral genome integration (the physical state of the viral DNA) is linked to a poorer prognosis in patients with viral DNA integration, similar to those with (HPV−) oral and pharyngeal squamous cell carcinoma (OPSCC) [136,139,141].

The integration of viral DNA is not a random event; it tends to occur in genomic regions that contain variations in copy numbers and structural variants. This can lead to dysregulated expression of host genes near the integration sites. Some of these dysregulated genes are associated with carcinoma progression, including *PD-L1*, *SOX2*, *TP63*, *FGFR3*, and *MYC* [139].

The expression of these viral oncogenes leads to the reprogramming of the epigenetic processes. This reprogramming induces two demethylases, KDM6A and KDM6B, which are enzymes that modify chromatin. They exert various effects, including activating HOX genes usually silenced by the Polycomb group (PcG). Additionally, E6 modulates miRs and other ncRNAs, which alters the regulation of genetic expression. The expression of DNA methyltransferases (*DNMT1* and *DNMT3A*) is upregulated, with HPV-16 E7 interacting with *DNMT1* [141].

The suppression of Rb prevents the normal cellular response to oncogene-induced senescence triggered by epigenetic reprogramming. This suppression avoids the typical cell cycle suppression caused by Rb. It causes the HPV-transformed cells to become dependent on the expression of HPV oncogenes, a phenomenon known as oncogene addiction [141].

The epigenetic reprogramming can occur through the HPV-16 E7-KDM6B axis, resulting in a dependence on the tumor suppressor protein p16 (also known as INK4A, one of the two cell cycle inhibitory proteins encoded by the *CDKN2A* gene). Once activated, p16 suppresses the activity of *CDK4* and *CDK6*, bestowing it with an oncogenic role [141].

In addition to the role of E6 and E7 in initiating tumorigenesis, the progression to carcinoma requires somatic alterations in the host cell genome. Interestingly, (HPV−) H&NSCC tend to have more copy-number alterations than the (HPV+) ones, indicating a higher level of genomic instability [141].

The most frequently mutated gene in (HPV–OPSCC is TP53, present in at least 75% of patients. In (HPV+) OPSCC, the TP53 mutation is found in a subset of heavy smokers due to the KRAS mutation, like what is seen in lung adenocarcinoma patients. Another pathway in (HPV+) OPSCC is through the off-target DNA editing activity associated with the *APOBEC* (APOBEC3 and APOBEC 3A enzymes), a family of cytidine deaminases whose role is to clear HPV DNA off the host genome [141].

The off-target DNA editing activity of the APOBEC is thought to have an oncogenic role when it fails to clear off the virus, leading to APOBEC-associated mutations, which are point mutations. However, copy number alterations can also occur. These mutations are seen in the PIK3CA helical domain and lead to the activation of the PIK3 signaling pathway, which seems to occur early in carcinogenesis, similar to the TP53 alteration. The most commonly mutated gene (20–30%) in (HPV+) OPSCC is the PI3K components (*PIK3CA*, *PIK3C2B*, and *PIK3R1*), and inactivating mutations in the negative regulators (*PTEN*, *TSC1*, or *TSC2*) have all been associated with more prolonged overall survival in (HPV+) OPSCC [141].

### Other Alterations

Among the genes involved in epidermal differentiation (TP63, KMT2D, NOTCH1, RIPK4, and NOTCH1), NOTCH1 mutations are associated with a significantly shorter overall survival duration, particularly in individuals with (HPV+) OPSCC [141].

While the role of the HIPPO pathway in facial development has not been described, this signaling pathway plays a role in brain development through YAP stabilization of SMAD1 in astrocyte specification [147].

## 7. Stem Cells in the Embryonic Period and Carcinoma

In our previous description, we have outlined the current understanding of genes that play a role in the development and progression of H&NSCC. This section will focus on the types of cells responsible for expressing these genes, specifically the stem cells that exhibit the stem cell phenotype, CD44 high, and aldehyde dehydrogenase [150,151]. These cells can be ESCs, germline stem cells, induced pluripotent stem cells (iPSCs) [149,152], ASCs serving as reserve cells for maintaining and repairing adult bodies, or CSCs responsible for driving malignant transformation [153].

Initially, embryonal carcinoma cells were the first isolated stem cells, but it was later discovered that stem cells are present in various tumors. PSCs and CSCs share several properties, including self-renewal, similar signaling pathways (Wnt or Notch), and specific surface markers such as CD133. Additionally, both types of cells exhibit a metabolic phenotype commonly seen in actively proliferating cells [154].

### 7.1. Embryonic Stem Cells

During the embryonic period, ESCs are derived from the pluripotent inner cell mass and are capable of self-renewal and multidirectional differentiation. This means that they can differentiate into all tissues and organs of adult animals, a capability that ASCs cannot achieve [152,155]. ESCs exist in two different states of pluripotency: a näive one that corresponds to an earlier and resembles cells from the pre-implantation embryo and a primed pluripotency stem cell found in the post-implantation epiblast [156].

In this period, bodies suffer from a head-to-tail (anteroposterior) formation. Driven by plastic axial progenitors capable of generating spinal cord neuroectoderm and pre-somitic/PM, both are considered precursors of spine and trunk muscles and are named neuro-mesodermal progenitors (NMP). They can be identified around the end of the gastrulation period or at the beginning of the somitogenesis; all the processes associated with the region’s posterior growth are parallel to the node-anterior primitive streak border and caudal lateral epiblast. Molecularly, the expression of pro-neural and pro-mesodermal transcription factors such as Sox2, brachyury, TBXT, Tbx6, and CDx2 characterizes them [157]. The interaction between lineage-specific transcription factors in an antagonistic way determines the differentiation of neural versus mesodermal cell types from NMP. These NMPs express Hox gene family members, activated during the posterior growth region. The posterior process is tied to acquired positional identity in the recently formed axial progenitor derivatives before they are located along the developing embryonic anteroposterior axis.

The NMP niches trigger the posteriorizing signaling pathways Wnt and FGF, and the transcription factor networks in the niches mentioned above potentiate Wnt/FGF activity within the posterior growth region during axis elongation through positive feedback. Wnt/FGF are tied to the progenitor maintenance and differentiation of the early neural and presomitic mesoderm cells [158,159,160,161,162]. Wnt and FGF drive Hox gene expression via cross-talk with the posteriorizing transcription factors CDX2 and TBXT [157,160].

Delta-Notch is another fundamental developmental signaling pathway in the posterior growth region/NMP niches. The pathway is activated by the interaction of receptors/ligands expressed in neighboring cells. In mammals, four receptors (NOTCH1-4) are linked to five Notch transmembrane ligands (DLL1, DLL3, DLL4, JAG2, and JAG2). NMPs express several Notch signaling molecules and BMP inhibitors from the underlying mesenchyme) and their immediate neural and mesodermal derivatives from late gastrulation [161] and embryonic axis elongation. Some of these molecules’ low expression or overexpression leads to severe posterior patterning alterations. These data suggest that the Notch pathway is a central component for NMP tissue specification and maintenance, but how it influences NMP ontogeny needs to be clarified [157]. The Notch attenuation during NMP induction deleteriously impacts activating pro-mesodermal transcription factors and global *HOX* gene activation. The Notch pathway can drive pro-mesodermal/HOX signaling, and some *HOX* genes in hESC-derived (human embryonic stem cells) NMPs could be mediated by Notch in a non-cell autonomous way [160].

The ESCs are dependent on the leukemia inhibitory factor (LIF) produced by the Trophoblast that keeps them away from differentiation through the LIFR and a co-receptor gp130 that activates the *JAK/STAT3* pathway; other transcription factors also play a role in maintaining the stemness capability; those are *BMP 4/2*, *Oct3/4*, *Nanog*, *Stat3 Sox2*, *c-Myc Esrrb*, *Klf4*, *Ronin*, *Tcl1*, *Tbx3,* and Rest. The combined action of *LIF* and *BMP4/2* completely blocks the differentiation of ESCs and maintains a high self-renewal capacity. Oct3/4 are the most crucial factors for pluripotency maintenance and differentiation to a multipotency state. The regulation of self-renewal is under the control of transcriptional factors, including *Nanog*, *STAT3*, *Oct3/4*, *Smad1*, *Sox2*, *n-Myc*, *c-Myc*, *Klf4*, *Esrrb*, *Zfx*, *E2f1*, and others [152,162].

Other modulators of stemness are the long intergenic non-coding RNA (lincRNAs) molecules of more than 200 nucleotides autonomously transcribed from the intergenic regions, but their role deserves more research [152].

MiRs also play a role in controlling pluripotency, self-renewal, and differentiation; studies on mice and human ESCs have demonstrated that the Dicer and Drosha knockdown resulted in defects in differentiation and proliferation. The control of these effects is mediated through the miR-290 cluster, the most abundantly expressed in ESCs and comprising up to 70% of miRs in undifferentiated ESCs. MiRs control differentiation through reducing expression of pluripotency factors; for example, we know that miR-296 represses Nanog; miR-134 and miR-470 target Nanog, Oct4, and Sox2; miR-200cmiR-203 and miR-183 repress Sox2 and Klf4; and miR-145 also represses Oct4, Sox2, and Klf4 [145].

### 7.2. Adult Stem Cells

ASCs can be in an active state or as quiescent cells (QASCs) that include hematopoietic stem cells (HSCs), skeletal muscle stem cells (MuSCs), neural stem cells, iPSCs, hair follicle stem cells, and mesenchymal stem cells such as fibro-adipogenic progenitors. Quiescence is a cell cycle state where the cell can be in and reversible exit when needed. This quiescent state is not quiescent, given that many signaling pathways must be active to keep this quiescence [149]. Even more, transcriptional networks are necessary to control ASC function and identity. Besides these, transcriptional networks and post-transcriptional regulatory mechanisms are required, given that ASC expresses genes crucial for SC activation, but protein synthesis and accumulation are repressed. These are reflected in the QASC features as reduced cell size, reduced mitochondrial number, lower levels of RNA transcription, low levels of proteins that are critical drivers of the cell cycle (cyclin A2, cyclin B1, cyclin E2), and high levels of cyclin-dependent kinase inhibitors (*CDKN1A*, *CDKN1B*, and *CDKN1C*); there are transcription factors that are expressed in different QASCs: MuSCs express PAX7 (which decreases when entering the active state), HFSCs express TCF3 and TCF4 (which are lost upon activation), and HSCs express HOXB5 and TCF15 [149].

Unlike the QASCs, ESC and iPSCs have an increase in mitochondrial number during ESC differentiation, the O2 consumption, and ATP production, with simultaneous lactate production decrease, conditions that suggest a switch in energy metabolism from glycolysis to OXPHOS (see below) to get proper cell differentiation; additionally, mitochondrial-based apoptosis can contribute to cell differentiation, and mtDNA mutations resulting in severe biochemical deficiency [154].

Other features are not studied here because that is out of the scope of this review (to see more detailed reviews on stem cells, see references [149,152,154,161], that help to maintain the quiescent state as epigenetic signatures (bivalent chromatin, expression of regulators of histone modifications), a low metabolic activity (see below), characterized by the fact that QASCs are in niches with low access to high levels of oxygen, they have low numbers of mitochondria. Hence, they rely on glycolysis and fatty acid oxidation [149].

Using extracellular flux assays, which measure the oxygen consumption rate that correlates with the OXPHOS or the extracellular acidification rate that correlates with the glycolytic activity, we can directly measure the metabolic activity of stem cells, but these values are not absolute quantifications for a better explanation of the tools for studying stem cell metabolism by Jackson BT and Finley LWS [163]. Maintaining stem cell pathways is fundamental in embryo development and during the adult life span in supporting stem cell populations, tissue homeostasis, and hematopoiesis [125]. Simultaneously, RA is involved in these shared onco-ontogenesis mechanisms.

### 7.3. Cancer Stem Cells

Almost 160 years have passed since the German pathologist Robert Virchow described similarities between embryonic and tumor tissues. But even in 1829, the gynecologic surgeon Joseph Récamier observed small cells (that appeared to be in an early developmental state) were present in some tumors [23].

The cancer stem cell theory was proposed in the late 1970s. The most recent clues to the feature of this CSC theory were described in 2008 by Prince and Ailles: (1) a small fraction of the cells of a tumor have tumorigenic capacities when transplanted into immunodeficient mice, (2) CSCs express distinctive surface markers, (3) the resulting tumors obtained from the CSCs contain tumorigenic and non-tumorigenic populations of the original tumor, and (4) CSCs can be serially transplanted [164]. This model of CSCs remains controversial because not all cancers express markers of CSCs.

CSCs represent a subset of the cells conforming to the tumor; those cells are capable of tumor initiation, stem cell-like self-renewal, proliferation, and aberrant differentiation to heterogeneous carcinoma cell types. CSCs have a series of features that make them unique, as they are at the top of the cells that form the cancerous tumor, are resistant to many anticancer therapies (through unique mechanisms to resist cell death as are modified anti-apoptotic machinery, increased pump activity, decreased cell division), and can take advantage of them [165], as we can see when pancreatic carcinoma cells are co-incubated with gemcitabine., where the CSCs increased the aggressive behavior and their number [23,151,165].

Conley et al. demonstrated that the use of tyrosine inhibitor kinases could duplicate the percentage of the CSCs in cancers that have developed resistance to these therapies [166]; this must be due to their slow replicating capacity [151], which drives metastatic phenotype through the induction of the EMT (through the snail one and snail two expressions; those are considered major EMT inducers). For the malignant growth, CSCs use similar signaling pathways and patterns of genetic expression as the ASCs do. As explained above, the aberrant regulation of the canonical and non-canonical WNT signaling has been demonstrated in many human cancers. One key feature of ASCs and CSCs is the chromosomal stability maintained through the telomere length conservation using the TERT subunit of the telomerase; a link between TERT and Wnt/β-catenin signaling has been described [156].

In H&NSCC, CSCs were first identified in 2007 by Prince et al. CSCs in H&NSCC express the following phenotypes: *CD44 high*, *CD24*, *CD98*, *CD133+*, *CD166 high*, *CD200* and aldehyde dehydrogenase (ALDH1A1), integrin-b1 and these are associated with tumorigenesis, metastasis [150,151] (where also markers of EMT as CMET, SLC3a2, TWIST, and SNAIL are expressed), treatment failure, resistance to chemo and radiotherapy; CD10 high levels are described in H&NSCC refractory to treatments associated with the expression of Oct3/4 and correspond to local recurrences, higher histologic tumor grade, and distant metastases. The expression of CD 90, together with CD 44, is associated with lymph node metastasis. Other markers are GRP78 and BMI1 [164,165,166,167]. Besides the side population cells, those cells that can form spheres in a medium contained in EFG have been identified as stem-like cells in H&NSCC [168]. All the CSCs can initiate tumors, but only a subset can initiate metastases. To date, two distinct populations of CSCs have been identified in H&NSCC. For those that express the *highCD44/highESA* epithelial phenotype, CSCs are proliferative, and those that express the mesenchymal highCD44/lowESA phenotype, which is migratory [150,168].

### 7.4. Metabolism in Stem Cell Niches

In 1920, Otto Warburg and Cori described how tumor cells grow through glycolysis with subsequent conversion of pyruvate to lactate despite being cultured in aerobic conditions. This phenomenon generates two molecules of ATP per glucose molecule. The enhanced glycolysis in aerobiosis was posteriorly named the Warburg effect [169,170,171,172].

Metabolic regulation must be sufficient to accomplish different functions that ASCs are committed to, such as sustained proliferation, quiescence, coping with cellular stress and cell death, niche requirements, and regulation of cell fate. To maintain a continuous proliferation rate, ASCs and CSCs must use metabolic pathways to acquire nutrients and meet cellular anabolic demands. For a more complete review of the hallmarks of stem cell metabolism, see the work of Jackson BT and Finley LWS [163].

PSC and carcinoma cells exhibit similar metabolic pathways directed to favor glycolysis under aerobiosis, through glycolysis in a shared metabolic behavior that ties proliferating cells and permits metabolic reactions to get the energy cells needed for cell division. In adult life, the conditions with the capacity to impact metabolic pathways and force cells to switch ways to produce energy can lead to a change in the cell status in pluripotency and differentiation, stem cells turning to cells to malignant behavior and cell transformation, suggesting that metabolic choices influencing the epigenetic program in these kinds of cells regulating stem cell physiology [154].

Conversely, normal adult somatic cells (except ASCs) have a different metabolic profile related to a state of terminal differentiation that is not involved in proliferation. Their metabolic environment is regulated in the context of the oxidative phosphorylation (OXPHOS) mitochondrial to produce energy and ATP [154].

The metabolic preference for the glycolytic way is a cellular tool in front of rapid proliferation, and it is related to the overexpression of glycolytic enzymes in normal stem cells, CSCs, and carcinoma cells associated with delayed cellular senescence in an environment of glycolysis and low OXPHOS [154,163].

The mechanisms related to pluripotency maintenance versus cell differentiation lead to a metabolic shift in the core pluripotency circuitry, which includes Oct4, Sox2, and Nanog. This shift converges with the signal transducer and activator of transcription 3 (STAT3), considered a master metabolic regulator that controls the OXPHOS to glycolysis shift [154,173].

Stem cells and CSCs have various mechanisms that involve genes that are early expressed during embryonic development. Wnt is a family of secreted signaling proteins with functions in embryogenesis, organogenesis, carcinoma cell development, and normal stem cell functions; it is considered a morphogen, functioning through concentration gradients with positional information to differentiate cells and tissues. 

Wnt, as a family of secreted signaling proteins, participates in a variety of functions during embryogenesis, organogenesis, malignant transformation, and stem cell metabolic processes, and the family proteins are considered a morphogen linked to concentration gradients with positional information for cellular or tissular differentiation; however, the extracellular distribution of Wnt proteins and their ligands, is poorly understood because Wnt proteins and ligands distribution involves biochemical and physical processes including diffusion with a random dispersion, dissociation and binding to cell surface molecules as the proteoglycans type heparan sulfate favoring their pleiotropic actions. Other morphogens acting together with Wnt are *BMP*, *TGF-β*, and *FGF* [174,175].

## 8. Discussion

The craniofacial region is highly complex, both anatomically and developmentally. Due to this complexity, malignant or benign tumors in this area are challenging to treat. 

Malignant transformation captures the embryonic developmental mechanisms and the adult wound repair ways to favor the malignant cells’ survival and proliferation [176]. It is a frequent situation that different cancers originate from transformed or deregulated immature progenitors, or ASCs, considered crucial cells for adult tissue remodeling and repair. Carcinoma genomic expression has the convergence of multiple embryonic mechanisms to enhance proliferation, migration, and metastasis. In epithelial cells, stem or immature cells are frequently more susceptible to suffering malignant transformation toward squamous cell carcinomas in a joint group of cancers in the head and neck, esophagus, lung, cervix uteri, and skin [176].

One implied mechanism is the lineage-dependent driver genes involved in the squamous development and stratification related to various molecular mechanisms driving tumor plasticity in squamous cancers, identifying novel drug targets and effective treatment options [176].

This paper focuses on that carcinoma reactivates embryonic pathways inactive in adult homeostatic tissues and that reactivation sustains malignant behavior. The similar characteristics between carcinoma and adult wound repair present analogies between malignant transformation and embryonic developmental mechanisms, injury repair, and tissue regeneration, suggesting that carcinoma captures the embryogenesis and wound repair mechanisms to support malignancy development [177].

The ASCs are the fundamental units used in postnatal tissue remodeling and repair. They take different ways, and some of them are constantly cycling (hematopoietic system, gastrointestinal epithelium, skin, and others), but it is essential not to consider CSCs as a new cell type with the latest characteristic cellular mechanisms and new molecular pathways; in fact, malignant cells are hijacking the mechanisms of the existing lineage programs used during embryonic period and by ASC and redirecting from a quiescent state to a stress-induced way during tumorigenesis [176]. This hijacking the microenvironment towards malignant transformation, tumor progression, invasion, and metastasis.

All these environmental modifications are strongly dependent on their own specific lineage primitive survival pathways based on their tissue of embryonic origin, such as basal cell carcinomas and the expression of the Shh signaling pathway or melanomas that express SOX family members molecules [177].

CSCs represent a cell population closely related to benign and malignant neoplasm triggers and progression in this context. They express reliable markers that could be useful in the early detection and characterization of H&NSCC and are tied to the expression of embryonic molecules.

The behavior of tumor cells during carcinogenesis (including progression, invasion, and migration) is similar to that of normal ESCs during embryogenesis. Several studies have shown a close relationship between various head and neck cancers and the expression of primary embryonic genes, such as the Wnt gene, which is involved in embryonic development. It is expected to find gene mutations in this pathway activated during tumor development in malignant cells; this is evident in the invasion and metastasis processes that can be paralleled to the process observed during the migration of NCCs and the invasion and metastasis of malignant cells [26,27]. The motility and invasion of cells are observed during embryogenesis, tissue remodeling, and in malignant tumors, which use similar mechanisms. These cells take on a flattened or mesenchymal shape with actin-rich cytoplasmic appendices, and they form adhesive interactions with the underlying surface, combined with contractile cell properties, to achieve cellular movement [178,179,180]. The selective activation of Rho and ARF gene families regulates the reorganization of the cytoskeleton required for amoeboid and mesenchymal movement.

A question remains to be answered: How do the CSCs “know” when to stop the developmental program to develop a tumor and not an embryonic structure?

One of the characteristics of developmental processes is the evolutive expression of antagonistic genes that inhibit or block the expression of genes with no more activity. Hence, those antagonist genes are not expressed in the CSCs, so the activity of the primordial genes continues, and the CSCs do not finish on a developed structure.

One of the embryonic molecules associated with carcinogenesis in H&NSCC is *PAX9* (paired box 9), a transcription factor of the *PAX* family genes functioning as a molecular transcriptional activator and repressor factor, with an oncogenic mechanism yet unknown [181], sometimes associated with upstream and others to downstream cascades in carcinoma development, as an example promoter hypermethylation promoter SNP, microRNA and inhibition upstream pathways in close relationship with molecules as *NOTCH* with *PAX9* silencing or downregulation, but too with gene amplification and epigenetic axis to upregulate *PAX9* expression; so *PAX9* contributes to carcinogenesis through dysregulation in transcriptional targets and molecular pathways in H&NSCC, esophageal, lung and cervical squamous cell cancers [181,182].

This transcriptional factor is relevant because it is primarily expressed during the embryonic period in immature tissues, such as the pharyngeal pouch endoderm, somites, and mesenchyme derived from the neural crest and distal limb buds [183]. *PAX9* has a primary role in craniofacial development. The loss of functioning has lethal results in murine models and the arrest of cleft palate formations and skeletal abnormalities. It is also involved in squamous cell differentiation and odontoblast differentiation of PSCs. Deregulation represents a close relationship with tumor triggering and malignant transformation through promoter hypermethylation with downregulation of squamous cells mediated by downregulation of differentiation and apoptosis. Conversely, activation of *PAX9* is linked to enhanced drug sensitivity during chemotherapy, turning a focus on possible carcinoma therapeutics [183,184].

The *WNT* signaling pathway, such as *PAX9*, is involved in developing 80% of epithelial-origin tumors in oral and maxillary sinus squamous cell cancers. The *WNT* way is essential in embryonic developmental regulatory ways and has a profound influence on malignant transformation triggering in H&NSCC [185]. In a recent report, *WNT* was analyzed in preparations from 85 patients with maxillary sinus squamous cell carcinoma and found a strong relation with malignantly transformed cells towards the presence of dysregulations of the *WNT* signaling ways, highlighting the molecule as a potential target molecule for potential therapy [185].

The HOX gene family is highly conserved and crucial for embryonic developmental mechanisms; it encodes transcription factors that regulate cell proliferation and differentiation. When HOX expression patterns are altered, it triggers a dysregulation toward abnormal cell proliferation and differentiation, especially HOXA13, which is involved in tumorigenesis and the development of squamous cancers. It even is related to carboplatin chemoresistance, associated with overexpressed HOX members, including *HOXA7*, *HOXA9*, *HOXA13*, *HOXB2*, *HOXB5*, *HOXB7*, *HOXB8*, *HOXB9*, and *HOXC9*. The EMT is also involved, where *HOXA10* controls malignant cell migration through the above-described mechanism in OSCC [186].

On the other hand, it has been observed the role of HPV and EBV in the development of H&NSCC. HPV-encoded E6 and ED7 genes are considered viral oncogenes with an essential role in malignant cell transformation [187], integrating the host gene genome. Their expression inactivates tumor suppressors p53 and pRb, even weakly, when the viruses are considered low-risk viruses. However, high and low risk are currently identified in H&NSCC. Another virus involved in malignant transformation is EBV, which is strongly associated with oral carcinoma progression by the action of the LMP-1 EBV-encoded oncogene when the human epithelial cells co-express Bcl-2. Recently published experimental papers have reported how the co-expression of HPV E6/E7, even in low-risk viruses, and EVB LMP molecules does not induce malignant transformation in squamous cells. Still, it permits the addition of somatic mutations by accumulated and increased DNA damage and suppression of DNA damage response, leading to immature squamous cells expressing precancerous alterations [187].

Despite the mechanisms above describing the high-risk HPV per se, the elevated O-linked GlcNAcylation (O-GlcNAc) linked to the enzyme O-Glc-NAc transferase (OGT) upregulates cell proliferation during transduction of E6 or E6/E7, reducing the cellular senesce through apoptotic ways in immature squamous cells [188].

Additionally, a phenomenon called “field cancerization” has been proposed in this region. This term was coined in 1953 to explain the high likelihood of local recurrences after treatments due to the presence of multiple primary independent tumors in the H&NSCC. This theory is supported by the finding that genetic changes in the mucosa surrounding the tumor (called patches) can be found in almost 35% of the analyzed oral and oropharyngeal tumors. These patches and the primary tumor had 50–100 carcinoma driver genes [143].

In 1996, the first genetic multi-step progression model for H&NSCC was proposed based on genetic characterization in the head and neck squamous epithelium. Changes such as loss of heterozygosity (LOH) during dysplasia and other genetic alterations in the late stages of oncogenesis were identified. This has led to the hypothetical “patch-field-tumor-metastasis-progression model”.

At this point, we cannot sustain or even refuse the inverse embryogenesis or reverse morphogenesis proposals totally because, in carcinoma development, there are no mature cells that involution from a developed tissue to an immature condition; ESCs and CSCs begin in their scenario an evolutive process. In tissues that are not transformed, the ESC ends the process with an anatomically developed structure; instead, the CSCs stop the evolutive process in an early phase. We do not know which mechanisms (miRNA, other RNA structures, post-transcriptional changes) the CSCs express the primary genes needed for survival, motility–migration, angiogenesis, and immortality.

## 9. Conclusions

Carcinoma and embryonic development share various phenotypical characteristics and upregulated molecular pathways. However, carcinoma is characterized by uncontrolled proliferation of malignant clones, abnormal angiogenesis, and invasive capacity. Common pathways shared include Wnt, Sonic Hedgehog (Shh), Hox genes, BMP family molecules, Notch, and proteases activated receptors [189]. Despite the multiple theories proposed to parallel embryogenesis to the carcinoma process, the remaining question is why the tumoral cells (CSC) do not express the totality of genes expressed during the total embryogenic process to finalize with a developed structure. The ontogenic carcinoma field theory proposed by Höckel could help to answer this question, but the molecular mechanism that controls the limited (to the initial phases of development) expression of genes in carcinoma is not quite well understood. Second, what is the role of miRNAs in stopping gene expression? Hence, more studies on the role of miRs in this field are warranted. Third, if oncogenic development breaks with embryonic development, how this can be explained by a single theory still needs to be answered. Meanwhile, the tools to speak of the onco-ontogeny theory of the H&NSCC are settled and wait to be completely understood.

## Figures and Tables

**Figure 1 ijms-25-09979-f001:**
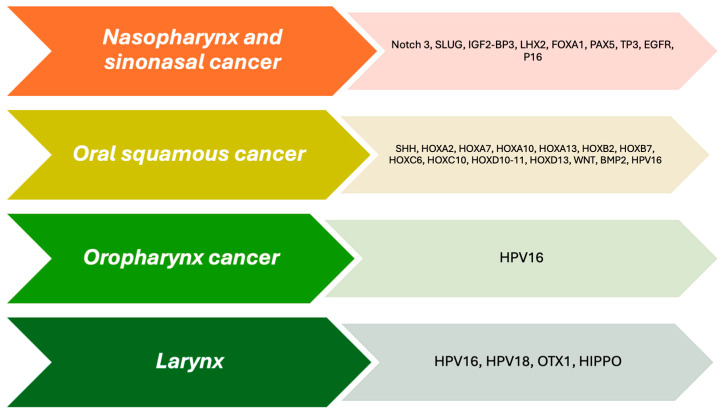
Factors related to malignant transformation in structures derivatives of the pharyngeal arches and pouches (to the genes and factors actions see description in Table 2).

**Table 1 ijms-25-09979-t001:** Transcription factors, signaling molecules, and embryonic structures activated during craniofacial development.

Transcription Factors, Signaling Molecules, and Embryonic Structures	Genes Activated	Mechanisms	Function
Neural crest cells (NCCs) (cranial ectoderm) *	*HOX* genes		Hindbrain segmentation (rhombomeres)
	NCC invade the pharyngeal pouches in three streams	1st stream cell migration 2nd stream cell migration	1st and 2nd rhombomeres cells invade 1st arch 4th rhombomere cells invade 6th arch
		3rd stream cell migration	6th and 7th rhombomeres cells invade rest of arches
Paraxial Mesoderm			
Anterior visceral endoderm (AVE)(pharyngeal pouches endoderm)	Stimulates facial ectoderm	Specialized migratory extraembryonic epithelial cells	Signaling center for multiple pattering events
	(Ectodermic frontonasal zone)	Induces neural crest mesenchyme precursors	To form facial bones
	Asymmetric expression of: *LEFTY I*, *CER I* and *HEX* genes	Regional organizing role of AVE	It explains that most of primary tumors in H&N originate in endoderm of pharyngeal pouches (squamous cell carcinoma)
Visceral endoderm	Asymmetric expression of *OTX2* and *DDK1*		Covers trophectoderm-derived extraembryonic ectoderm
	*Nodal y MAPK*	In response to SMAD2 phosphorylation	Transform in AVE cells
Visceral distal endoderm	Interaction with Nodal and *MAPK* and *SMAD2* phosphorylation	Induces differentiation	as AVE
Forebrain and associated structures	Influenced by genes *LHX1*, *EMX1*, *EMX2*, *OTX1*, and *OTX2*	Induces signals on the prechordal mesoderm or the AVE	Pouch (endodermal origin)Arch (mesenchymal origin)Grooves (ectodermal covering)
First pharyngeal arch	Independent of *HOX* genes and RA Dependent on the action *OTX2*	Inductor signals to form: 1st aortic arch (maxillary artery), trigeminal (V) development	Formation of muscles of mastication, tensor tympany, veli palatini tensor digastric (anterior belly), middle ear ossicles (malleus, incus), sphenomadibular ligament, Meckel cartilage, tympanic ring
Pharyngeal pouches 2–4	RA in an ascendent gradient in craniocaudal sense	Inductor signals to form: 2nd aortic arch (hyoid artery, stapedial artery), facial nerve (VII) development	Origin for muscles: facial expression, stapedial, stylohyoid, digastric (posterior belly), stapes, styloid process, stylohyoid ligament, hyoid (lesser horn, and part of body)
		3rd aortic arch: internal carotid artery and glossopharyngeal nerve (IX)) development	Stylopharyngeus muscle, hyoid (greater horn and part of the body)
		4th aortic arch (right subclavian and aortic arteries), vagus nerve (X) development	Pharyngeal and laryngeal muscles, laryngeal cartilages

* See Figure 1 to complete the information.

## Data Availability

This study did not create or analyze new data, and data sharing does not apply to this article.

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
