# Peer review of "Onco-Ontogeny of Squamous Cell Cancer of the First Pharyngeal Arch Derivatives"

_ijms, 2024, doi:10.3390/ijms25189979_

Round 1

Reviewer 1 Report

Comments and Suggestions for Authors

Introduction 

Lane 75 

It will be more better if the author include EB virus information on the nasopharyngeal cancer.

Lane 87 

It also will be more better if the include the distant metastasis situation.

2. Molecular embryology of head and neck, 

There are so many related genes of this term. I feel that the author should split after 2.2 paragraph because the after 2.2 paragraph, they are related to the cancer development.

So from 2.3 terms, I feel the author would change the 3. terms, like as 3.1 EMX1 and EMX2, ~

From above reasons, 3. Head and neck cancer: regions and subregions should be changed to 4. Then after the same pattern.

This is a very descriptive paper. It provides a lot of information.

However, I felt that there was very little discussion of a clear relationship or mechanism between the genes described in the paper and the actual genes responsible for carcinogenesis.

It would be wonderful if the authors could provide more information on the relationship between the genes and the actual carcinogenesis, and if they could expand the discussion with references to make the paper more meaningful.

Author Response

Introduction 

Lane 75 

It will be more better if the author include EB virus information on the nasopharyngeal cancer.

Answer

We added EBV information and two new references in line 76 to 77 of the document,

Lane 87 

It also will be more better if the include the distant metastasis situation.

Answer

We add a phrase in lines 89 to 91 concerning the distant metastasis situation and the new references.

  1. Molecular embryology of head and neck, 

There are so many related genes of this term. I feel that the author should split after 2.2 paragraph because the after 2.2 paragraph, they are related to the cancer development.

So from 2.3 terms, I feel the author would change the 3. terms, like as 3.1 EMX1 and EMX2, ~

From above reasons, 3. Head and neck cancer: regions and subregions should be changed to 4. Then after the same pattern.

Answer

We followed your recommendation and renamed the titles of the suggested numbers

This is a very descriptive paper. It provides a lot of information.

However, I felt that there was very little discussion of a clear relationship or mechanism between the genes described in the paper and the actual genes responsible for carcinogenesis.

It would be wonderful if the authors could provide more information on the relationship between the genes and the actual carcinogenesis, and if they could expand the discussion with references to make the paper more meaningful.

Answer

We reorganize the Discussion section, introducing lines 1204 to 1239, talking about how the malignant transformation captures the embryonic developmental mechanisms and wound repair processes to drive to malignant tissue sustainability and how transduction factors and genes originally expressed during embryo development are reactivated during the onco-ontogeny process in a tumor hijacking of developmental processes using form Sonic Hedgehog and SOX signaling pathways to inhibit the normal cell apoptotic ways.

In paragraphs 1261 to 1317, we discuss how other molecules, such as the NOTCH, PAX, WNT, and HOX gene family and their interaction with DNA viruses, can participate in malignant transformation using embryonic methods to assure tumor survival through different metabolic routes.

With these issues, we discuss the molecular implications that could be clinically more relevant by their impact on the evolution of malignant diseases in the head and neck region.

Reviewer 2 Report

Comments and Suggestions for Authors

The manuscript attempts to draw a parallel between genes associated with head and neck development and their potential role in the persistence or onset of cancers in this region. The topic is both highly interesting and original. However, due to the complexity of the subject matter, the text would benefit from a more didactic approach to make it less confusing and more accessible. I have provided some suggestions for reorganization in this initial review. Following this, I will conduct a more detailed examination of the theoretical content discussed. Additionally, please include figures that summarize the main findings of the review.

Introduction

- Lines 70-71: These lines are out of context, as age is not the sole determinant in the genesis of these cancers. I suggest beginning the introduction at line 72.

- Lines 95-98: The challenges of late diagnosis and treatment resistance are far more critical than age alone as determining factors in treatment. This section should be rewritten to reflect that.

- Line 99: What analysis tools are being referred to? Please provide a reference.

- Lines 95-105: The most crucial points in this section are in lines 99, 100, and 101. The other lines contain information that does not significantly contribute to the primary objective of this paper. A general paragraph discussing the primary etiology of each region might be more beneficial.

-Lines 106-111: Is this the only evidence presented? This paragraph is the most important in the introduction. Additional points motivating this review should be included here.

Overview Section

- Include item 2 within a section titled "Overview"

- Lines 145-146: These suppositions should be removed at this stage of the review.

- Ensure that all abbreviations are spelled out the first time they are mentioned. Gene names should always be italicized.

Development of the Face Section

- Create a figure illustrating the critical moments and key signaling pathways in facial development.

- Subclassify the topics under item 2.2. Begin with a brief introductory paragraph and then create a subtopic for EMX genes and others. Currently, this section is confusing. The authors should reorganize this section by separating genes (with each gene family as a subtopic) from signaling pathways.

- Item 2.3 should become an independent section following the "HNSCC section”. Subdivide it similarly.

Reorganization Suggestion

To enhance clarity and readability, I propose the following reorganization of the text:

1. Introduction

2. Embryology overview

2.1. Molecular development of the head and neck

2.2. Molecular development of the first arch

2.3. Stem cells in the embryonic period (move this item here)

2.4. Key genes related to head and neck development

3. Head and neck cancer by anatomical location (remove the term "regions and subregions")

   - Clearly outline the etiological factors for each region. A table detailing this information for each region would be beneficial.

4. Genes associated with embryonic development and their role in HNSCC

4.1. HNSCC associated with tobacco and alcohol

4.2. HNSCC associated with viral oncogenesis (including EBV+ nasopharyngeal cancers)

5. Role of cancer stem cells in HNSCC

6. Parallels between embryonic development and malignant cells

7. Conclusion

Discussion Section

- The discussion should be reorganized by integrating the corresponding paragraphs into sections 3, 4, 5, or 6.

Figures

- A figure illustrating the genes associated with development that continue to influence the development of head and neck cancers would make the article more didactic and easier to understand.

Comments on the Quality of English Language

 Minor editing of English language required.

Author Response

Comments and Suggestions for Authors

The manuscript attempts to draw a parallel between genes associated with head and neck development and their potential role in the persistence or onset of cancers in this region. The topic is both highly interesting and original. However, due to the complexity of the subject matter, the text would benefit from a more didactic approach to make it less confusing and more accessible. I have provided some suggestions for reorganization in this initial review. Following this, I will conduct a more detailed examination of the theoretical content discussed. Additionally, please include figures that summarize the main findings of the review.

Introduction

- Lines 70-71: These lines are out of context, as age is not the sole determinant in the genesis of these cancers. I suggest beginning the introduction at line 72.

Answer

We agree with you concerning age and its role in cancer development; it is not a unique factor, however, in our clinical experience, more than 70% of our cases are in patients over 60 years old; by this clinical observation, we are considering the aging processes interacting with the embryonic processes reactivation we talk about both factors as part of our introduction

- Lines 95-98: The challenges of late diagnosis and treatment resistance are far more critical than age alone as determining factors in treatment. This section should be rewritten to reflect that.

Answer

We agree with you concerning the late diagnosis and treatment resistance; however, in our clinical experience are, factors such as nutritional conditions in patients the most important determinants in survival and treatment response, particularly in a clinical setting with limited access to target therapies by budget restrictions.

The analysis of our own patient populations, recently published, gives us the analytical and scientific support to sustain our point of view (Sat-Muñoz D, Martínez-Herrera BE, González-Rodríguez JA, Gutiérrez-Rodríguez LX, Trujillo-Hernández B, Quiroga-Morales LA, Alcaráz-Wong AA, Dávalos-Cobián C, Solórzano-Meléndez A, Flores-Carlos JD, Rubio-Jurado B, Salazar-Páramo M, Carrillo-Nuñez GG, Gómez-Sánchez E, Nava-Zavala AH, Balderas-Peña LM. Phase Angle, a Cornerstone of Outcome in Head and Neck Cancer. Nutrients. 2022 Jul 24;14(15):3030. doi: 10.3390/nu14153030. PMID: 35893884; PMCID: PMC9330539.; Martínez-Herrera BE, Gutiérrez-Rodríguez LX, Trujillo-Hernández B, Muñoz-García MG, Cervantes-González LM, José Ochoa LL, González-Rodríguez JA, Solórzano-Meléndez A, Gómez-Sánchez E, Carrillo-Nuñez GG, Salazar-Páramo M, Nava-Zavala AH, Velázquez-Flores MC, Nuño-Guzmán CM, Mireles-Ramírez MA, Balderas-Peña LM, Sat-Muñoz D. Phase Angle in Head and Neck Cancer: A Sex-Differential Analysis from Biological and Clinical Behavior to Health-Related Quality of Life. Biomedicines. 2023 Jun 12;11(6):1696. doi: 10.3390/biomedicines11061696. PMID: 37371791; PMCID: PMC10296649.). We add the references concerning our findings to support this part of the document, which is now located between lines 95 and 102 and supported by references 14 and 15.

- Line 99: What analysis tools are being referred to? Please provide a reference.

Answer

We add references 16 and 17(Rauf S, Ullah S, Abid MA, Ullah A, Khan G, Khan AU, Ahmad G, Ijaz M, Ahmad S, Faisal S. A computational study of gene expression patterns in head and neck squamous cell carcinoma using TCGA data. Future Sci OA. 2024 Dec 31;10(1):2380590. doi: 10.1080/20565623.2024.2380590. Epub 2024 Aug 14. PMID: 39140365; PMCID: PMC11326450.).

- Lines 95-105: The most crucial points in this section are in lines 99, 100, and 101. The other lines contain information that does not significantly contribute to the primary objective of this paper. A general paragraph discussing the primary etiology of each region might be more beneficial.

-Lines 106-111: Is this the only evidence presented? This paragraph is the most important in the introduction. Additional points motivating this review should be included here.

Answer

We modified some sections in the introduction and they are highlighted in blue, we think that enhances the introduction structure

Overview Section

- Include item 2 within a section titled "Overview"

- Lines 145-146: These suppositions should be removed at this stage of the review.

- Ensure that all abbreviations are spelled out the first time they are mentioned. Gene names should always be italicized.

Answer

We reorganized the introduction and considered that the structure had been enhanced

Development of the Face Section

- Create a figure illustrating the critical moments and key signaling pathways in facial development.

- Subclassify the topics under item 2.2. Begin with a brief introductory paragraph and then create a subtopic for EMX genes and others. Currently, this section is confusing. The authors should reorganize this section by separating genes (with each gene family as a subtopic) from signaling pathways.

- Item 2.3 should become an independent section following the "HNSCC section”. Subdivide it similarly.

Reorganization Suggestion

To enhance clarity and readability, I propose the following reorganization of the text:

  1. Introduction
  2. Embryology overview

2.1. Molecular development of the head and neck

2.2. Molecular development of the first arch

2.3. Stem cells in the embryonic period (move this item here)

2.4. Key genes related to head and neck development

  1. Head and neck cancer by anatomical location (remove the term "regions and subregions")

   - Clearly outline the etiological factors for each region. A table detailing this information for each region would be beneficial.

  1. Genes associated with embryonic development and their role in HNSCC

4.1. HNSCC associated with tobacco and alcohol

4.2. HNSCC associated with viral oncogenesis (including EBV+ nasopharyngeal cancers)

  1. Role of cancer stem cells in HNSCC
  2. Parallels between embryonic development and malignant cells
  3. Conclusion

Answer

We have reorganized the complete structure of the document, trying to find coincidences between reviewers. 

Discussion Section

- The discussion should be reorganized by integrating the corresponding paragraphs into sections 3, 4, 5, or 6.

Answer

We added at least ten paragraphs to the discussion section, considering the points of view of both reviewers

Round 2

Reviewer 2 Report

Comments and Suggestions for Authors

The authors have addressed some of my concerns from the previous review. However, the manuscript still presents serious issues in organization, typographical errors, and a lack of standardization in abbreviations. For the article to be published, it needs to be carefully reviewed by all authors, line by line. Additionally, the text is excessively long and confusing.

Here are some suggestions for a second review, emphasizing points that were not addressed in the previous revision:

1. Introduction: Since this is a review article focused on head and neck squamous cell carcinoma (HNSCC), and considering the readers of this journal who may use the article as a reference, the authors should address the most common characteristics across the general population, regardless of the clinical experience of the group. It is acceptable for the authors to mention and reference their clinical experience, but other factors must also be highlighted. The classic association of HNSCC with elderly patients is changing, especially with the emergence of new epidemiological groups, including younger patients. Many readers, especially students, should not be led to believe that age is the primary factor associated with the tumor's onset without a clear mention of other factors such as alcohol, tobacco, viruses, UV radiation, and genetic factors. In patients over 60 years of age, those who smoked and drank will have a significantly higher chance of developing cancer than those who never smoked. Some cancers are more associated with age due to the accumulated exposure to certain carcinogens rather than cellular senescence, which has been recently included in the hallmarks of cancer.

2. I agree with the authors on the importance of nutritional factors, but again, beyond clinical experience, patients face other challenges in treatment, such as toxicities related to radiotherapy and chemotherapy, resistance to these treatments, late diagnosis, and the need for prevention, early diagnosis, and population awareness. These points should also be mentioned. Remember, this is a literature review article, and the introduction cannot be biased.

3. When the authors mention "which results in a higher occurrence of age-related mutational changes," it is necessary to clarify the consequence of this for increasing treatment complexity. The text is unclear on this point and needs to be revised.

4. Subtitles: Remove references to tables and figures from the section subtitles.

5. Gene formatting: Genes should be italicized, as requested in the previous review.

6. Figure 1: The caption for Figure 1 should be more comprehensive and explanatory.

7. Article length: The article is excessively long and tedious. Suggestion 1: Review the manuscript to remove information that may not be essential. Suggestion 2: Reorganize the discussion paragraphs throughout the review, as previously suggested, so that discussion points are addressed as the topics arise after the introduction. This will make the reading more interesting and fluid.

8. References: Sections 2 and 2.1 are almost entirely based on references 20 and 21. It is necessary to add other references to diversify the sources.

9. Section 2.2 structure: As mentioned in the previous review, it is important to revise the structure of the text. It begins by discussing EMX genes without a dedicated topic, then introduces a topic for Wnt signaling, and later does not have a separate topic for LIM genes. There is a mix of genes and pathways, making the text confusing.

10. Use of abbreviations: Retinoic acid is mentioned 16 times in the article. In the first mention, the authors use the abbreviation RA, but later repeat "retinoic acid" in several instances. Please, reread the entire manuscript and reorganize the abbreviations and technical terms. For example, the explanation of the abbreviation EMX only appears on the eighth mention in the text, which is just one example among many others. On line 1313, the authors revisit the abbreviation H&NSCC. I ask all 15 authors involved to carefully review the text. As it stands, the publication is not recommended.

11. Reorganization: When I suggested creating an overview, it was to avoid having to return to the normal development of the head and neck on page 11 when we are already reading about cancer.

12. Section organization: Item 3 should be included within item 4, in the regions where the studies were conducted. After discussing the genes related to development in cancer, there is another topic with more information, now divided by areas. Why not reorganize and unify this? Simplify. The article is too long. The authors could include the salivary glands as an additional region if necessary.

13. Line 926 - Replace "Now, I'd like to focus…" with "In this section, we will focus on the types of cells...".

14. Item 7 should come immediately after the introduction. It does not make sense for this "historical introduction" to be at the end of the article, especially since it underscores the importance of conducting a review like this.

15. Standardize the use of "squamous cell cancer" or "squamous cell carcinoma" throughout the manuscript.

Comments on the Quality of English Language

There are numerous typographical and punctuation errors throughout the text.

Author Response

Comments and Suggestions for Authors

The authors have addressed some of my concerns from the previous review. However, the manuscript still presents serious issues in organization, typographical errors, and a lack of standardization in abbreviations. For the article to be published, it needs to be carefully reviewed by all authors, line by line. Additionally, the text is excessively long and confusing.

Here are some suggestions for a second review, emphasizing points that were not addressed in the previous revision:

  1. Introduction: Since this is a review article focused on head and neck squamous cell carcinoma (HNSCC), and considering the readers of this journal who may use the article as a reference, the authors should address the most common characteristics across the general population, regardless of the clinical experience of the group. It is acceptable for the authors to mention and reference their clinical experience, but other factors must also be highlighted. The classic association of HNSCC with elderly patients is changing, especially with the emergence of new epidemiological groups, including younger patients. Many readers, especially students, should not be led to believe that age is the primary factor associated with the tumor's onset without a clear mention of other factors such as alcohol, tobacco, viruses, UV radiation, and genetic factors. In patients over 60 years of age, those who smoked and drank will have a significantly higher chance of developing cancer than those who never smoked. Some cancers are more associated with age due to the accumulated exposure to certain carcinogens rather than cellular senescence, which has been recently included in the hallmarks of cancer.

ANSWER:

In lines 70-72, I added the reference to data published by Johnson, Barsuk, and Sung. They describe the increase in new cases linked to aging and a sustained pattern in the disease's epidemiology behavior.

In the second paragraph, we describe the role of alcohol, tobacco, and viruses in the etiology of H&NSCC (lines 73-78).

In lines 85 and 86, we mentioned how, despite so many cases diagnosed in subjects over 50 years old, the epidemiological data trends to diminish the incidence partially by a reduction in tobacco use.

The role of UV radiation is essential in skin cancer of the head and neck, but in mucosal H&NSCC, UV radiation has not been demonstrated to have a role. It is necessary to clarify this point.

  1. I agree with the authors on the importance of nutritional factors, but again, beyond clinical experience, patients face other challenges in treatment, such as toxicities related to radiotherapy and chemotherapy, resistance to these treatments, late diagnosis, and the need for prevention, early diagnosis, and population awareness. These points should also be mentioned. Remember, this is a literature review article, and the introduction cannot be biased.

ANSWER

The paragraph mentions the co-morbidities related to parallel issues to solve in patients, and these co-morbidities turn into a complex scenario where frequently the patients are unsuitable for treatment but it is not the intention to mention the challenges faced by patients during the treatment.

We reproduce the paragraph to show to the reviewer and editor the complete structure (lines 96-103):

“The treatment of these types of cancers is very complex due to the anatomical region involved, requiring a multidisciplinary approach involving medical oncologists, surgical oncologists, maxillofacial surgeons, radiation oncologists, plastic surgeons, nutritionists, psycho-oncologists, and neurosurgeons [15]. This means that the treatment needs to be multimodal. However, it is complicated by the fact that most patients are around 60 years old on average, which results in other associated pathologies such as hypertension, diabetes, and malnutrition[16–18]. These additional health issues could make a patient unsuitable for treatment.”

  1. When the authors mention "which results in a higher occurrence of age-related mutational changes," it is necessary to clarify the consequence of this for increasing treatment complexity. The text is unclear on this point and needs to be revised.

ANSWER

We eliminated the phrase “which results in a higher occurrence of age-related mutational changes” in line 101.

  1. Subtitles: Remove references to tables and figures from the section subtitles.

ANSWER

We move the reference below in the body of the paragraph (line 649).

  1. Gene formatting: Genes should be italicized, as requested in the previous review.

ANSWER

We italicized the names of the genes throughout the document.

  1. Figure 1:The caption for Figure 1 should be more comprehensive and explanatory.

ANSWER

We change the caption for Figure 1

“Factors related to malignant transformation in structures derivatives of the pharyngeal arches and pouches (to the genes and factors actions see description in table 2)

  1. Article length: The article is excessively long and tedious. Suggestion 1: Review the manuscript to remove information that may not be essential. Suggestion 2: Reorganize the discussion paragraphs throughout the review, as previously suggested, so that discussion points are addressed as the topics arise after the introduction. This will make the reading more interesting and fluid.

ANSWER:

It is not our intention to be tedious but, the field of this article is not entirely known, and there are a lot of genes that need to be more investigated, even their association in the onco-ontogeny of the first pharyngeal arch; we reorganized and rewrote some sections, and we think that changes can turn it more readable. 

  1. References: Sections 2 and 2.1 are almost entirely based on references 20 and 21. It is necessary to add other references to diversify the sources.

ANSWER:

Unfortunately, the most comprehensive references are those authors we refer to in the molecular embryology of the head and neck (Carstens and Carlson). We look for papers from them that could add to this comprehensive review.

  1. Stower, M.J.; Srinivas, S. Heading Forwards: Anterior Visceral Endoderm Migration in Patterning the Mouse Embryo. Philos. Trans. R. Soc. B Biol. Sci. 2014, 369, 20130546, doi:10.1098/rstb.2013.0546.
  2. Klimova, L.; Antosova, B.; Kuzelova, A.; Strnad, H.; Kozmik, Z. Onecut1 and Onecut2 Transcription Factors Operate Downstream of Pax6 to Regulate Horizontal Cell Development. Dev. Biol. 2015, 402, 48–60, doi:10.1016/j.ydbio.2015.02.023.
  3. Shioi, G.; Hoshino, H.; Abe, T.; Kiyonari, H.; Nakao, K.; Meng, W.; Furuta, Y.; Fujimori, T.; Aizawa, S. Apical Constriction in Distal Visceral Endoderm Cells Initiates Global, Collective Cell Rearrangement in Embryonic Visceral Endoderm to Form Anterior Visceral Endoderm. Dev. Biol. 2017, 429, 20–30, doi:10.1016/j.ydbio.2017.07.004.
  4. Robertson, E.J. Dose-Dependent Nodal/Smad Signals Pattern the Early Mouse Embryo. Semin. Cell Dev. Biol. 2014, 32, 73–79, doi:10.1016/j.semcdb.2014.03.028.
  5. Williams, A.L.; Bohnsack, B.L. What’s Retinoic Acid Got to Do with It? Retinoic Acid Regulation of the Neural Crest in Craniofacial and Ocular Development. Genes. N. Y. N 2000 2019, 57, e23308, doi:10.1002/dvg.23308.

All of them mentioned in the section 3 and 3.1

  1. Section 2.2 structure: As mentioned in the previous review, it is important to revise the structure of the text. It begins by discussing EMX genes without a dedicated topic, then introduces a topic for Wnt signaling, and later does not have a separate topic for LIM genes. There is a mix of genes and pathways, making the text confusing.

ANSWER:

In section 3.2, we added a paragraph to clarify why we mentioned diverse genes at the beginning of the section lines 265-269).

  1. Use of abbreviations:Retinoic acid is mentioned 16 times in the article. In the first mention, the authors use the abbreviation RA, but later repeat "retinoic acid" in several instances. Please, reread the entire manuscript and reorganize the abbreviations and technical terms. For example, the explanation of the abbreviation EMX only appears on the eighth mention in the text, which is just one example among many others. On line 1313, the authors revisit the abbreviation H&NSCC. I ask all 15 authors involved to carefully review the text. As it stands, the publication is not recommended.

ANSWER:

We corrected the abbreviations throughout the text RA and H&NSCC.

  1. Reorganization:When I suggested creating an overview, it was to avoid having to return to the normal development of the head and neck on page 11 when we are already reading about cancer.

ANSWER

We reorganized and introduced changes to make the article easily readable.

  1. Section organization: Item 3 should be included within item 4, in the regions where the studies were conducted. After discussing the genes related to development in cancer, there is another topic with more information, now divided by areas. Why not reorganize and unify this? Simplify. The article is too long. The authors could include the salivary glands as an additional region if necessary.

ANSWER

We renamed the sections according to the reviewer's global suggestions. .

  1. Line 926 - Replace "Now, I'd like to focus…" with "In this section, we will focus on the types of cells...".

ANSWER

We replaced the phrase: “Now, I'd like to focus with” with “In this section, we will focus on the types of cells”

  1. Item 7 should come immediately after the introduction. It does not make sense for this "historical introduction" to be at the end of the article, especially since it underscores the importance of conducting a review like this.

ANSWER

We renumbered the sections, and now section 7 is renamed section 2, according to the reviewer’s suggestion.

  1. Standardize the use of "squamous cell cancer" or "squamous cell carcinoma" throughout the manuscript.

ANSWER

We standardized the terms cancer to carcinoma throughout carcinoma.

Comments on the Quality of English Language

There are numerous typographical and punctuation errors throughout the text.

ANSWER

We request the style correction through the MDPI platform

Submission Date:

16 July 2024

Date of this review

27 Aug 2024 02:36:27